# The Reliability of OKRidge Method in Solving Sparse Ridge Regression Problems

**Xiyuan Li    Youjun Wang    Weiwei Liu**[*]
School of Computer Science, Wuhan University
National Engineering Research Center for Multimedia Software, Wuhan University
Institute of Artificial Intelligence, Wuhan University
Hubei Key Laboratory of Multimedia and Network Communication Engineering, Wuhan University
Lee_xiyuan@outlook.com, youjunw1208@gmail.com, liuweiwei863@gmail.com

## Abstract

Sparse ridge regression problems play a significant role across various domains. To solve sparse ridge regression, [1] recently proposes an advanced algorithm, Scalable Optimal $K$-Sparse Ridge Regression (OKRidge), which is both faster and more accurate than existing approaches. However, the absence of theoretical analysis on the error of OKRidge impedes its large-scale applications. In this paper, we reframe the estimation error of OKRidge as a Primary Optimization (**PO**) problem and employ the Convex Gaussian min-max theorem (CGMT) to simplify the **PO** problem into an Auxiliary Optimization (**AO**) problem. Subsequently, we provide a theoretical error analysis for OKRidge based on the **AO** problem. This error analysis improves the theoretical reliability of OKRidge. We also conduct experiments to verify our theorems and the results are in excellent agreement with our theoretical findings.

## 1 Introduction

Sparse Ridge Regression (SRR) has achieved notable success across various machine learning applications, including statistics [2], signal processing [3], dynamical systems [4, 5], and others. In this paper, we are interested in addressing the following $k$-sparse linear regression problem with additive noise:

$$\boldsymbol{y} = \boldsymbol{X}\boldsymbol{\beta}^* + \boldsymbol{\epsilon} \quad \text{with} \ \ \|\boldsymbol{\beta}^*\|_0 \le k, \tag{1}$$

where $\boldsymbol{\beta}^* \in \mathbb{R}^d$ represents the "true" weight parameter, $\boldsymbol{X} = (\boldsymbol{x}_1, \boldsymbol{x}_2, \cdots, \boldsymbol{x}_n)^\top \in \mathbb{R}^{n \times d}$ is the input measurement matrix, $\boldsymbol{y} = (y_1, y_2, \cdots, y_n)^\top \in \mathbb{R}^n$ is the real output responses, $\boldsymbol{\epsilon} = (\epsilon_1, \epsilon_2, \cdots, \epsilon_n)^\top \in \mathbb{R}^n$ is the noise vector, $k \in \mathbb{Z}^+$ specifies the maximum number of nonzero elements for the model, $\|\cdot\|_0$ denotes the number of nonzero elements of the given vector. Moreover, the entries of $\boldsymbol{X}$ are drawn i.i.d. from $\mathcal{N}(0,1)$; the entries of $\boldsymbol{\epsilon}$ are drawn i.i.d. from $\mathcal{N}(0, \sigma^2)$; and we assume $\frac{k}{d}$ is a constant and $\lim_{d \to \infty} \frac{n(d)}{d} = \delta \in (0, 1)$.

The formulation (1) represents a black box model where $\boldsymbol{\beta}^*$ is fixed. Given $\boldsymbol{X}$ and $\boldsymbol{y}$, to determine the target vector $\boldsymbol{\beta}^*$, the most basic method is solving the following $k$-Sparse Ridge Regression Optimization ($k$-SRO), as outlined by [1, 6, 7]:

$$\min_{\boldsymbol{\beta}} \|\boldsymbol{y} - \boldsymbol{X}\boldsymbol{\beta}\|_2^2 + \lambda\|\boldsymbol{\beta}\|_2^2 \quad \text{s.t.} \quad \|\boldsymbol{\beta}\|_0 \le k, \tag{2}$$

where $\lambda > 0$ is a regularizer parameter, and $\|\cdot\|_2$ denotes the Euclidean norm. Our paper focuses on the worst-case scenario $\|\boldsymbol{\beta}^*\|_0 = k$. This $k$-SRO is different from the traditional ridge regression

---

[*]Corresponding author: Weiwei Liu (liuweiwei863@gmail.com).

38th Conference on Neural Information Processing Systems (NeurIPS 2024).

due to the constraint of $k$-sparse structure for $\boldsymbol{\beta}$. The $k$-SRO problem (2) is NP-hard, and is more challenging in the presence of highly correlated features [8].

Two main types of algorithms are commonly employed for solving $k$-SRO problem (2): heuristic algorithms [9, 10] and optimal algorithms [11]. However, heuristic algorithms lack the ability to assess the solution quality, while the optimal algorithms are slow. In order to rapidly solve $k$-SRO problem (2) while ensuring solution optimality, [1] introduces a highly efficient method called OKRidge. Therefore, a complete algorithm of OKRidge, including how to choose hyper-parameters can be seen in the original paper [1]. OKRidge substitutes $k$-SRO problem (2) with an unconstrained optimization on a novel tight lower bound. The experiment results in [1] show that OKRidge is superior to heuristic algorithms, optimal algorithms, and existing mixed-integer programming (MIP) formulations solved by the commercial solver Gurobi. Nevertheless, the absence of theoretical error analysis for OKRidge impedes its scalability in practical applications.

In this paper, we provide theoretical error analysis for OKRidge utilizing the framework of the CGMT [12]. Specifically, we propose another novel tight lower bound $\mathcal{L}_{\text{OKRidge}}(\boldsymbol{\beta})$ to replace $k$-SRO problem (2):

$$\mathcal{L}_{\text{OKRidge}}(\boldsymbol{\beta}) := \|\boldsymbol{y} - \boldsymbol{X}\boldsymbol{\beta}\|_2^2 + \lambda \text{SumTop}_k(\boldsymbol{\beta} \odot \boldsymbol{\beta}), \tag{3}$$

where $\odot$ denotes Hadamard product, and $\text{SumTop}_k(\cdot)$ represents the summation of the largest $k$ elements of a given vector. The tight lower bound (3) is equivalent to that proposed by [1]. Thus, $\mathcal{L}_{\text{OKRidge}}(\boldsymbol{\beta})$ can replace the objective function of OKRidge. It is noteworthy that our proposed regularizer, defined as $\gamma(\boldsymbol{\beta}) = \text{SumTop}_k(\boldsymbol{\beta} \odot \boldsymbol{\beta})$, differs from any previously proposed instances by [12]. Then, the optimal solution obtained by OKRidge is

$$\hat{\boldsymbol{\beta}} = \arg\min_{\boldsymbol{\beta}} \mathcal{L}_{\text{OKRidge}}(\boldsymbol{\beta}). \tag{4}$$

[1] utilizes $\hat{\boldsymbol{\beta}}$ as the estimate of $\boldsymbol{\beta}^*$ in problem (1). By combining formulations (1) and (3), the estimation error of OKRidge can be obtained through the following normalized optimization problem:

$$\min_{\boldsymbol{w}} \frac{1}{\sqrt{n}} \left[ \|\boldsymbol{X}\boldsymbol{w} - \boldsymbol{\epsilon}\|_2^2 + \lambda \text{SumTop}_k\big((\boldsymbol{w} + \boldsymbol{\beta}^*) \odot (\boldsymbol{w} + \boldsymbol{\beta}^*)\big) \right], \tag{5}$$

where $\boldsymbol{w} := \boldsymbol{\beta} - \boldsymbol{\beta}^*$ is a random variable with randomness from the random variables $\boldsymbol{X}$ and $\boldsymbol{\epsilon}$, and the estimation error can be measured by $\|\boldsymbol{w}\|_2$. Subsequently, we transform the optimization (5) into a **PO** problem about the error of OKRidge, using the Fenchel-Moreau theorem [13]. Then, we employ the CGMT framework to substitute the complex **PO** problem with a simplified **AO** problem. Finally, we present the theoretical error analysis of OKRidge based on the **AO** problem. Our theoretical results focus on the Normalized Squared Error (NSE) of OKRidge and can be summarized as:

$$\lim_{d \to \infty} \lim_{\sigma \to 0} \text{NSE} \xrightarrow{P} \Delta(\hat{\lambda}), \tag{6}$$

where $\text{NSE} := \|\hat{\boldsymbol{\beta}} - \boldsymbol{\beta}^*\|_2^2/\sigma^2$, and $\Delta(\hat{\lambda})$ is a function of $\lambda$. These theoretical results indicate that if the regularizer parameter $\lambda$ used in OKRidge is constant, the NSE limit of OKRidge is also fixed. Moreover, $\hat{\boldsymbol{\beta}}$ learned by OKRidge is reliable to estimate $\boldsymbol{\beta}^*$, due to $\lim_{d \to \infty} \lim_{\sigma \to 0} \|\hat{\boldsymbol{\beta}} - \boldsymbol{\beta}^*\|_2 \xrightarrow{P} 0$. The comprehensive experiments of OKRidge on real-world examples were conducted by the NeurIPS 2023 paper [1] (see Figure 3 and Appendix H in [1]), which demonstrates that the error of OKRidge tends to zero. Our analysis explains the experimental phenomenon observed in [1], strengthens the theoretical underpinnings of OKRidge, and provides theoretical reliability for its broad application.

We also conduct numerical experiments to validate our theorems. The findings demonstrate that the NSE converges to a fixed constant determined by $\lambda$, aligning excellently with our theoretical predictions.

## 1.1 Outline

The structure of the remaining sections in this paper is as follows. Section 2 provides a review of related work. Section 3 offers background information on OKRidge, CGMT, and basic concepts. Section 4 introduces an alternative tight lower bound for the objective function of OKRidge. In Section 5, we convert the estimation error of OKRidge into a **PO** problem and simplify it into an **AO** problem using CGMT. Subsequently, an estimation error analysis of OKRidge based on the **AO** problem is conducted. Section 6 presents the experimental results. Finally, we conclude with a summary in Section 7. Additionally, the limitaion and impact of our work are detailed in Appendix A

## 2 Related work

### 2.1 Heuristic and Optimal Methods

Heuristic methods approximate solutions to optimization problems based on practical experience [14], including ensemble methods [15], swapping features [16], greedy methods [17], etc. While heuristic methods are fast, they often become trapped in local minima, and their solution quality cannot be assessed due to the absence of a lower bound on performance. Optimal methods aim to precisely solve sparse regression problems, such as the big-M method [18], the conditional-value-at-risk (CVaR) approach [19], big-M free mixed integer second order conic (MISOC) method [19], and so on. However, exact optimal methods are slow, particularly for large instances, to achieve near-optimality [20, 21]. To address the limitations of heuristic and optimal methods, [1] proposes an efficient approximation algorithm, OKRidge. Experimental results in [1] demonstrate that OKRidge outperforms heuristic algorithms, optimal algorithms, and existing MIP formulations solved by the commercial solver Gurobi.

### 2.2 Lower Bound Methods

Lower bound methods are capable of solving the NP-hard $k$-SRO problems. Several algorithms utilize the lower bound method, such as SOS1 formulation [18], big-M formulation [18], Subset Selection CIO method [22], and others. However, the SOS1 formulation lacks scalability in high dimensions, the big-M formulation is sensitive to hyperparameters, and the Subset Selection CIO method runs slowly. Recently, the perspective formulation [6, 23, 24] has been employed to induce a convex relaxed lower bound that is easier to solve. Building upon the perspective formulation, [1] proposes a novel lower bound used as the objective function for the OKRidge method.

### 2.3 Normalized Squared Error

NSE, defined as $\|\hat{\boldsymbol{\beta}} - \boldsymbol{\beta}^*\|_2^2 / \sigma^2$, serves as a natural measure of the estimation error. NSE is an important indicator in signal-to-noise ratio scenes [25, 26]. Bounds on NSE have been derived by [27, 28]. Additionally, [29] is the first to precisely formulate the limiting behavior of NSE. These studies primarily consider a Gaussian sensing matrix $\boldsymbol{X}$ and utilize the Approximate Message Passing (AMP) framework for analysis [30, 31]. These achievements motivate us to utilize NSE for evaluating the estimation error of the OKRidge method.

## 3 Preliminary

### 3.1 Relaxed Transformation of $k$-SRO

According to [1], the $k$-SRO problem (2) can be reformulated as the following optimization problem:

$$\min_{\boldsymbol{\beta}} \mathcal{L}_{\text{ridge}}(\boldsymbol{\beta}), \quad \text{s.t.} \begin{cases} (1-z_j)\beta_j = 0, \ j = 1, 2, \cdots, d, \\ \sum_{j=1}^d z_j \leq k, \ z_j \in \{0, 1\}, \end{cases} \tag{7}$$

where $\boldsymbol{z} = (z_1, z_2, \cdots, z_n)^\top \in \mathbb{R}^d$, and $\mathcal{L}_{\text{ridge}}(\boldsymbol{\beta}) := \|\boldsymbol{y} - \boldsymbol{X}\boldsymbol{\beta}\|_2^2 + \lambda \sum_{j=1}^d \beta_j^2$. This problem (7) remains NP-hard under the sparsity constraint [32]. Existing methods such as SOS1, big-M, or the perspective formulation do not leverage the $k$-sparse structure of the problem. [1] develops a novel method, OKRidge, to preserve the special structure through the following relaxed transformation.

By employing the perspective formulation [33, 34] and the Fenchel conjugate [35], [1] transforms the problem (7) to a new perspective optimization problem:

$$\min_{\boldsymbol{\beta}, \boldsymbol{z}} \max_{\boldsymbol{c}} \mathcal{L}_{\text{ridge}}^{\text{Fenchel}}(\boldsymbol{\beta}, \boldsymbol{z}, \boldsymbol{c}), \quad \text{s.t.} \sum_{j=1}^d z_j \leq k, \ z_j \in \{0, 1\}. \tag{8}$$

where $\boldsymbol{c} = (c_1, c_2, \cdots, c_n) \in \mathbb{R}^d$, and $\mathcal{L}_{\text{ridge}}^{\text{Fenchel}}(\boldsymbol{\beta}, \boldsymbol{z}, \boldsymbol{c}) := \|\boldsymbol{y} - \boldsymbol{X}\boldsymbol{\beta}\|_2^2 + \lambda \sum_{j=1}^d (\beta_j c_j - \frac{c_j^2}{4} z_j)$. This transformation does not change the optimal solution of the problem (7) [1, 35], indicating that problem (7) can be replaced by problem (8). To efficiently solve problem (8), [1] further relaxs

the binary constraint $\{0, 1\}$ to the interval $[0, 1]$, ultimately yielding the following relaxed convex optimization problem:

$$\min_{\boldsymbol{\beta}, \boldsymbol{z}} \max_{\boldsymbol{c}} \mathcal{L}_{\text{ridge}}^{\text{Fenchel}}(\boldsymbol{\beta}, \boldsymbol{z}, \boldsymbol{c}), \quad \text{s.t.} \sum_{j=1}^{d} z_j \leq k, \ z_j \in [0, 1]. \tag{9}$$

According to problem (7), to preserve the special sparse structure of $\boldsymbol{\beta}$, we always have $\beta_j = 0$ if $z_j = 0$. Directly solving the min-max problem (9) is computationally challenging. [1] utilizes the relaxed problem (9) to obtain a tight lower bound for the problem (8), where the lower bound corresponds to the objective function of OKRidge.

## 3.2 The Convex Gaussian Min-max Theorem

The CGMT framework, introduced by [28], has been utilized to analyze the performance of solutions to non-smooth regularized convex optimization problems. It has achieved significant success in various practical applications, including regularized logistic regression [36], max-margin classifiers [37], adversarial training [38, 39] and others. These achievements inspire us to apply the CGMT framework to analyze the NSE of the OKRidge method.

CGMT originates from Gordon's Gaussian Min-max Theorem (GMT) [40], which provides probabilistic bounds on the optimal cost of **PO** problem via a simpler **AO** problem. CGMT further tightens the bounds under convexity assumptions. According to GMT, [41] introduces the following asymptotic sequence and notation.

**Definition 3.1** (GMT admissible sequence). The sequence $\big\{ \boldsymbol{G}^{(d)}, \boldsymbol{g}^{(d)}, \boldsymbol{h}^{(d)}, \mathcal{S}_{\boldsymbol{w}}^{(d)}, \mathcal{S}_{\boldsymbol{u}}^{(d)}, \psi^{(d)} \big\}_{d \in \mathbb{N}}$ indexed by $d$, with $\boldsymbol{G}^{(d)} \in \mathbb{R}^{n \times d}$, $\boldsymbol{g}^{(d)} \in \mathbb{R}^n$, $\boldsymbol{h}^{(d)} \in \mathbb{R}^d$, $\mathcal{S}_{\boldsymbol{w}}^{(d)} \subset \mathbb{R}^d$, $\mathcal{S}_{\boldsymbol{u}}^{(d)} \subset \mathbb{R}^n$, $\psi^{(d)} : \mathcal{S}_{\boldsymbol{w}}^{(d)} \times \mathcal{S}_{\boldsymbol{u}}^{(d)} \to \mathbb{R}$ and $n = n(d)$, is said to be admissible if, for each $d \in \mathbb{N}$, $\mathcal{S}_{\boldsymbol{w}}^{(d)}$ and $\mathcal{S}_{\boldsymbol{u}}^{(d)}$ are compact sets and $\psi^{(d)}$ is continuous on its domain. Onwards, we will drop the superscript $(d)$ from $\boldsymbol{G}^{(d)}$, $\boldsymbol{g}^{(d)}$, $\boldsymbol{h}^{(d)}$.

A sequence $\big\{ \boldsymbol{G}^{(d)}, \boldsymbol{g}^{(d)}, \boldsymbol{h}^{(d)}, \mathcal{S}_{\boldsymbol{w}}^{(d)}, \mathcal{S}_{\boldsymbol{u}}^{(d)}, \psi^{(d)} \big\}_{d \in \mathbb{N}}$ defines a sequence of min-max problems

$$\Phi^{(d)}(\boldsymbol{G}) := \min_{\boldsymbol{w} \in \mathcal{S}_{\boldsymbol{w}}^{(d)}} \max_{\boldsymbol{u} \in \mathcal{S}_{\boldsymbol{u}}^{(d)}} \boldsymbol{u}^\top \boldsymbol{G} \boldsymbol{w} + \psi^{(d)}(\boldsymbol{w}, \boldsymbol{u}), \tag{10}$$

$$\phi^{(d)}(\boldsymbol{g}, \boldsymbol{h}) := \min_{\boldsymbol{w} \in \mathcal{S}_{\boldsymbol{w}}^{(d)}} \max_{\boldsymbol{u} \in \mathcal{S}_{\boldsymbol{u}}^{(d)}} \|\boldsymbol{w}\|_2 \boldsymbol{g}^\top \boldsymbol{u} + \|\boldsymbol{u}\|_2 \boldsymbol{h}^\top \boldsymbol{w} + \psi^{(d)}(\boldsymbol{w}, \boldsymbol{u}). \tag{11}$$

Importantly, the formulation (10) is called Primary Optimization (**PO**) and the formulation (11) is called Auxiliary Optimization (**AO**). Additionally, let $\boldsymbol{w}_\Phi^{(d)}(\boldsymbol{G})$ denote the optimal minimizer of **PO** problem (10), and $\boldsymbol{w}_\phi^{(d)}(\boldsymbol{g}, \boldsymbol{h})$ denote the optimal minimizer of **AO** problem (11). Define $\upsilon^{(d)} : \mathcal{S}_{\boldsymbol{w}}^{(d)} \to \mathbb{R}$ as follows,

$$\upsilon^{(d)}(\boldsymbol{w}; \boldsymbol{g}, \boldsymbol{h}) := \max_{\boldsymbol{u} \in \mathcal{S}_{\boldsymbol{u}}^{(d)}} \|\boldsymbol{w}\|_2 \boldsymbol{g}^\top \boldsymbol{u} + \|\boldsymbol{u}\|_2 \boldsymbol{h}^\top \boldsymbol{w} + \psi^{(d)}(\boldsymbol{w}, \boldsymbol{u}). \tag{12}$$

Clearly, $\phi^{(d)}(\boldsymbol{g}, \boldsymbol{h}) := \min_{\boldsymbol{w} \in \mathcal{S}_{\boldsymbol{w}}^{(d)}} \upsilon^{(d)}(\boldsymbol{w}; \boldsymbol{g}, \boldsymbol{h})$. For a sequence of random variables $\{\mathcal{X}^{(d)}\}_{d \in \mathbb{N}}$ and a constant $c \in \mathbb{R}$, $\mathcal{X}^{(d)} \xrightarrow{P} c$ denotes convergence in probability, i.e. $\forall \epsilon > 0$, $\lim_{d \to \infty} \mathbb{P}\big(|\mathcal{X}^{(d)} - c| > \epsilon\big) = 0$. Based on the GMT admissible sequence and the notation introduced above, we present the CGMT below.

**Theorem 3.2** (CGMT [12]). *Let* $\big\{ \boldsymbol{G}^{(d)}, \boldsymbol{g}^{(d)}, \boldsymbol{h}^{(d)}, \mathcal{S}_{\boldsymbol{w}}^{(d)}, \mathcal{S}_{\boldsymbol{u}}^{(d)}, \psi^{(d)} \big\}_{d \in \mathbb{N}}$ *be a GMT admissible sequence as in Definition 3.1, for which additionally the entries of* $\boldsymbol{G}$, $\boldsymbol{g}$, $\boldsymbol{h}$ *are drawn i.i.d. from* $\mathcal{N}(0, 1)$. *Let* $\Phi^{(d)}(\boldsymbol{G})$, $\phi^{(d)}(\boldsymbol{g}, \boldsymbol{h})$ *be the optimal costs, and,* $\boldsymbol{w}_\Phi^{(d)}(\boldsymbol{G})$, $\boldsymbol{w}_\phi^{(d)}(\boldsymbol{g}, \boldsymbol{h})$ *the corresponding optimal minimizers of the* **PO** *and* **AO** *problems in (10) and (11). The following three statements hold*

*(i) For any* $d \in \mathbb{N}$ *and* $c \in \mathbb{R}$,

$$\mathbb{P}\big(\Phi^{(d)}(\boldsymbol{G}) < c\big) \leq 2\mathbb{P}\big(\phi^{(d)}(\boldsymbol{g}, \boldsymbol{h}) \leq c\big).$$

(ii) *For any $d \in \mathbb{N}$. If $\mathcal{S}_{\boldsymbol{w}}^{(d)}$, $\mathcal{S}_{\boldsymbol{u}}^{(d)}$ are convex, and, $\psi^{(d)}(\cdot, \cdot)$ is convex-concave on $\mathcal{S}_{\boldsymbol{w}}^{(d)} \times \mathcal{S}_{\boldsymbol{u}}^{(d)}$, then, for any $\mu \in \mathbb{R}$ and $t > 0$,*

$$\mathbb{P}\big(|\Phi^{(d)}(\boldsymbol{G}) - \mu| > t\big) \leq 2\mathbb{P}\big(|\phi^{(d)}(\boldsymbol{g}, \boldsymbol{h}) - \mu| > t\big).$$

(iii) *Assume the conditions of (ii) hold for all $d \in \mathbb{N}$. Let $\|\cdot\|$ denote some norm in $\mathbb{R}^d$ and recall (12). If, there exist constants (independent of d) $\kappa^*$, $\alpha^*$ and $\tau > 0$ such that*

(a) $\phi^{(d)}(\boldsymbol{g}, \boldsymbol{h}) \xrightarrow{P} \kappa^*$,

(b) $\|\boldsymbol{w}_{\phi}^{(d)}(\boldsymbol{g}, \boldsymbol{h})\| \xrightarrow{P} \alpha^*$,

(c) *with probability one in the limit $d \to \infty$*

$$\Big\{ \upsilon^{(d)}(\boldsymbol{w}; \boldsymbol{g}, \boldsymbol{h}) \geq \phi^{(d)}(\boldsymbol{g}, \boldsymbol{h}) + \tau\big(\|\boldsymbol{w}\| - \boldsymbol{w}_{\phi}^{(d)}(\boldsymbol{g}, \boldsymbol{h})\big)^2, \forall \boldsymbol{w} \in \mathcal{S}_{\boldsymbol{w}}^{(d)} \Big\},$$

*then,*

$$\|\boldsymbol{w}_{\Phi}^{(d)}(\boldsymbol{G})\| \xrightarrow{P} \alpha^*. \tag{13}$$

Theorem 3.2 indicates that, if the optimal cost $\phi(\boldsymbol{g}, \boldsymbol{h})$ of (11) concentrates to some value $\mu$, the same holds true for $\Phi(\boldsymbol{G})$ of (10). Furthermore, under appropriate additional assumptions, the optimal solutions of the **AO** and **PO** problems are also closely related by $\|\boldsymbol{w}_{\Phi}(\boldsymbol{G})\| = \|\boldsymbol{w}_{\phi}(\boldsymbol{g}, \boldsymbol{h})\|$, as $n \to \infty$. This suggests that, within the CGMT framework, a challenging **PO** problem can be replaced with a simplified **AO** problem, from which the optimal solution of the **PO** problem can be accurately inferred [12]. Subsequently, we rewrite the lower bound of problem (9) in the form of **PO** problem (10) and analyze the minimizer of the simplified **AO** problem instead.

### 3.3 Basic Concept

Suppose $f : \mathbb{R}^d \to \mathbb{R}$ and $\boldsymbol{u}, \boldsymbol{v} \in \mathbb{R}^d$, the Fenchel conjugate of $f$ is defined as $f^*(\boldsymbol{u}) = \sup_{\boldsymbol{v}} \boldsymbol{v}^\top \boldsymbol{u} - f(\boldsymbol{v})$. Additionally, $f^*$ is always convex and lower semi-continuous. According to the Fenchel-Moreau theorem [13], if $f$ is convex and continuous, we have $f(\boldsymbol{v}) = \sup_{\boldsymbol{u}} \boldsymbol{u}^\top \boldsymbol{v} - f^*(\boldsymbol{u})$. In this paper, we utilize the following conjugate pairs

$$f(\boldsymbol{v}) = \|\boldsymbol{v}\|_2^2 \leftrightarrow f^*(\boldsymbol{u}) = \frac{\|\boldsymbol{u}\|_2^2}{4}. \tag{14}$$

If $\gamma(\cdot) : \mathbb{R}^d \to \mathbb{R}$ is a convex function of $\boldsymbol{\beta}$, the subdifferential of $\gamma(\cdot)$ at $\boldsymbol{\beta}^*$ is the set of vectors: $\partial \gamma(\boldsymbol{\beta}^*) = \{\boldsymbol{s} \in \mathbb{R}^d | \gamma(\boldsymbol{\beta}^* + \boldsymbol{u}) \geq \gamma(\boldsymbol{\beta}^*) + \boldsymbol{s}^\top \boldsymbol{u}\}$. According to [13], $\partial \gamma(\boldsymbol{\beta}^*)$ is nonempty, convex and compact. Given $\boldsymbol{h} \in \mathbb{R}^d$, we define $\text{dist}(\boldsymbol{h}, \partial \gamma(\boldsymbol{\beta}^*)) = \min_{\boldsymbol{s} \in \partial \gamma(\boldsymbol{\beta}^*)} \|\boldsymbol{h} - \boldsymbol{s}\|_2$. Then, the Gaussian squared distance corresponding to the scaled subdifferential is defined as $D(\tau) := D_{\partial \gamma(\boldsymbol{\beta}^*)}(\tau) := \mathbb{E}_{\boldsymbol{h}}\big[\text{dist}^2(\boldsymbol{h}, \tau \partial \gamma(\boldsymbol{\beta}^*))\big]$, where $\tau > 0$. Suppose $C(\tau) = -\frac{\tau}{2}\frac{\partial D(\tau)}{\partial \tau}$, $\lim_{d \to \infty} \frac{n}{d} \to \delta \in (0, 1)$, $\lim_{d \to \infty} \frac{D(\tau)}{n} \to \bar{D}(\tau) \in (0, 1)$, $\lim_{d \to \infty} \frac{C(\tau)}{n} \to \bar{C}(\tau)$. Based on the Gaussian squared distance, we define a $map$ function:

$$\text{map}(\tau) := \frac{1 - \bar{C}(\tau) - \bar{D}(\tau)}{\sqrt{1 - \bar{D}(\tau)}}, \tau > 0. \tag{15}$$

We denote $\lambda_{map}$ as the solution of $\text{map}(\tau) - \lambda/2 = 0$. Since $map(\tau)$ depends on $\gamma(\cdot)$ and $\boldsymbol{\beta}^*$, when the form of $\gamma(\cdot)$ and the value of $\boldsymbol{\beta}^*$ are determined, the $\lambda_{map}$ is fixed.

## 4   Tight Lower Bound in OKRidge

In this section, we utilize problem (9) to derive another novel lower bound for problem (8), serving as used as the objective function of OKRidge. Our lower bound is equivalent to the tight lower bound provided by [1]. Specifically, [1] eliminates the parameter $\boldsymbol{c}$ in problem (9) by setting the gradient of $\boldsymbol{\beta}$ to $\boldsymbol{0}$, while we eliminate the parameter $\boldsymbol{c}$ by setting the gradient of $\boldsymbol{c}$ to $\boldsymbol{0}$. These two methods

are equivalent due to the independence and convexity of $c$ and $\beta$. Given any $\beta$ and $z$, the optimality condition for $c$ in problem (9) is taking $\partial \mathcal{L}_{\text{ridge}}^{\text{Fenchel}}(\beta, z, c)/\partial c = 0$. Therefore, we have

$$\frac{\partial \mathcal{L}_{\text{ridge}}^{\text{Fenchel}}(\beta, z, c)}{\partial c} = \beta - \frac{\text{diag}(z)c}{2} = 0, \tag{16}$$

$$\Rightarrow c_j = \begin{cases} \rho \in \mathbb{R} & \text{, if } z_j = 0, \\ \frac{2\beta_j}{z_j} & \text{, if } z_j \neq 0, \end{cases} \tag{17}$$

where $\text{diag}(z)$ is a diagonal matrix with $z$ on the diagonal. Inspired by this optimality condition, we present the following theorem.

**Theorem 4.1.** *If we define the parameter $c$ as (16), the problem (9) is equivalent to the following optimization problem:*

$$\min_{\beta, z} \mathcal{L}_{\text{ridge}}^{\text{saddle}}(\beta, z), \quad s.t. \sum_{j=1}^{d} z_j \leq k, \ z_j \in [0, 1], \tag{18}$$

*where*

$$\mathcal{L}_{\text{ridge}}^{\text{saddle}}(\beta, z) := \|y - X\beta\|_2^2 + \lambda \sum_{j=1, z_j \neq 0}^{d} \frac{\beta_j^2}{z_j}. \tag{19}$$

The proof of Theorem 4.1 follows Theorem 3.1 of [1] and is included in Appendix B for completeness. Following the approach by [1], we can approximately solve the problem (18) while still obtaining a feasible lower bound. We define a new function $\mathcal{L}(\beta)$ as:

$$\mathcal{L}(\beta) = \min_{z} \mathcal{L}_{\text{ridge}}^{\text{saddle}}(\beta, z), \quad s.t. \sum_{j=1}^{d} z_j \leq k, \ z_j \in [0, 1]. \tag{20}$$

For any $\beta$, $\mathcal{L}(\beta)$ serves as a valid lower bound for problem (7). We should choose $z$ such that this lower bound $\mathcal{L}(\beta)$ is tight.

**Theorem 4.2.** *The function $\mathcal{L}(\beta)$ defined in Equation (20) is lower bounded by*

$$\mathcal{L}(\beta) \geq \|y - X\beta\|_2^2 + \lambda SumTop_k(\beta \odot \beta). \tag{21}$$

*where $\odot$ is Hadamard product, and $SumTop_k(\cdot)$ denotes the summation of the largest $k$ elements of a given vector.*

The proof of Theorem 4.2 can be seen in Appendix C. Based on (9), (18) and (20), the tight lower bound (21) is equivalent to the one provided by [1], as both are derived through equivalent processes. OKRidge solves the original $k$-sparse problem (7) using this tight lower bound (21) as its objective function. If we define

$$\mathcal{L}_{\text{OKRidge}}(\beta) := \|y - X\beta\|_2^2 + \lambda SumTop_k(\beta \odot \beta),$$

OKRidge solves $k$-SRO problem (2) with

$$\min_{\beta} \mathcal{L}_{\text{OKRidge}}(\beta), \tag{22}$$

where we obtain $\mathcal{L}_{\text{OKRidge}}$ of formulation (3). So far, we transform the constrained $k$-SRO problem (2) into the unconstrained optimization problem (22). Let

$$\hat{\beta} = \text{argmin}_{\beta} \ \mathcal{L}_{\text{OKRidge}}(\beta),$$

OKRidge regards $\hat{\beta}$ as the estimation of $\beta^*$ in problem (1). Next, we apply CGMT to analyze the error $\|\hat{\beta} - \beta^*\|_2^2$ for OKRidge.

# 5 The Error Analysis for OKRidge

## 5.1 From PO to AO

As discussed in Section 4, the estimation error of OKRidge is characterized by $\|\hat{\boldsymbol{\beta}} - \boldsymbol{\beta}^*\|_2^2$. Taking formulation (1) into the properly normalized objective (22), OKRidge (22) can be equivalently transformed to the following optimization:

$$\min_{\boldsymbol{\beta}} \frac{1}{\sqrt{n}} \left[ \|\boldsymbol{X}(\boldsymbol{\beta} - \boldsymbol{\beta}^*) + \boldsymbol{\epsilon}\|_2^2 + \lambda \mathrm{SumTop}_k(\boldsymbol{\beta} \odot \boldsymbol{\beta}) \right]. \tag{23}$$

The crucial step is to convert (23) into a **PO** problem within the framework of CGMT. We introduce the new variable $\boldsymbol{w} := \boldsymbol{\beta} - \boldsymbol{\beta}^*$ and apply the Fenchel-Moreau theorem (14) to formulation (23),

$$\frac{1}{\sqrt{n}} \left[ \|\boldsymbol{X}\boldsymbol{w} - \boldsymbol{\epsilon}\|_2^2 + \lambda \mathrm{SumTop}_k\big((\boldsymbol{w} + \boldsymbol{\beta}^*) \odot (\boldsymbol{w} + \boldsymbol{\beta}^*)\big) \right]$$

$$= \max_{\boldsymbol{u}} \frac{1}{\sqrt{n}} \left[ \boldsymbol{u}^\top \boldsymbol{X}\boldsymbol{w} - \boldsymbol{u}^\top \boldsymbol{\epsilon} - \frac{\|\boldsymbol{u}\|_2^2}{4} + \lambda \mathrm{SumTop}_k\big((\boldsymbol{w} + \boldsymbol{\beta}^*) \odot (\boldsymbol{w} + \boldsymbol{\beta}^*)\big) \right], \tag{24}$$

where $\boldsymbol{w} \in \mathbb{R}^d, \boldsymbol{u} \in \mathbb{R}^n$. Based on (10) and (24), the **PO** problem corresponding to the estimation error of OKRidge is

$$\Phi_{\mathrm{OKRidge}}(\boldsymbol{X}) = \min_{\boldsymbol{w}} \max_{\boldsymbol{u}} \frac{1}{\sqrt{n}} \big( \boldsymbol{u}^\top \boldsymbol{X}\boldsymbol{w} + \psi(\boldsymbol{w}, \boldsymbol{u}) \big), \tag{25}$$

where

$$\psi(\boldsymbol{w}, \boldsymbol{u}) := -\boldsymbol{u}^\top \boldsymbol{\epsilon} - \frac{\|\boldsymbol{u}\|_2^2}{4} + \lambda \mathrm{SumTop}_k\big((\boldsymbol{w} + \boldsymbol{\beta}^*) \odot (\boldsymbol{w} + \boldsymbol{\beta}^*)\big). \tag{26}$$

Since the entries of $\boldsymbol{X}$ are drawn i.i.d. from $\mathcal{N}(0, 1)$, to replace the challenging **PO** problem (25) with a simplified **AO** problem through CGMT, $\psi(\boldsymbol{w}, \boldsymbol{u})$ should be a convex-concave function. The following Lemma illustrates that the $\psi(\boldsymbol{w}, \boldsymbol{u})$ satisfies the conditions of Theorem 3.2.

**Lemma 5.1.** *Suppose $\psi(\boldsymbol{w}, \boldsymbol{u})$ is defined as in formulation (26). Then, $\psi(\boldsymbol{w}, \boldsymbol{u})$ is convex-concave function.*

The proof of Lemma 5.1 can be seen in Appendix D. Define

$$\gamma(\boldsymbol{\beta}) := \mathrm{SumTop}_k(\boldsymbol{\beta} \odot \boldsymbol{\beta}).$$

Because the **PO** problem (25) satisfies the assumptions of CGMT, we transform it to the following **AO** problem:

$$\phi_{\mathrm{OKRidge}}(\boldsymbol{g}, \boldsymbol{h}) = \min_{\boldsymbol{w}} \max_{\boldsymbol{u}} \frac{1}{\sqrt{n}} \left[ \|\boldsymbol{w}\|_2 \boldsymbol{g}^\top \boldsymbol{u} + \|\boldsymbol{u}\|_2 \boldsymbol{h}^\top \boldsymbol{w} - \boldsymbol{\epsilon}^\top \boldsymbol{u} - \frac{\|\boldsymbol{u}\|_2^2}{4} + \lambda \gamma(\boldsymbol{\beta}^* + \boldsymbol{w}) \right]$$

$$= \min_{\boldsymbol{w}} \max_{\boldsymbol{u}} \frac{1}{\sqrt{n}} \left[ (\|\boldsymbol{w}\|_2 \boldsymbol{g} - \boldsymbol{\epsilon})^\top \boldsymbol{u} + \|\boldsymbol{u}\|_2 \boldsymbol{h}^\top \boldsymbol{w} - \frac{\|\boldsymbol{u}\|_2^2}{4} + \lambda \gamma(\boldsymbol{\beta}^* + \boldsymbol{w}) \right], \tag{27}$$

where the entries $\boldsymbol{g}, \boldsymbol{h}$ are drawn i.i.d. from $\mathcal{N}(0, 1)$, due to the property of $\boldsymbol{X}$. Suppose $\boldsymbol{w}_{\Phi_{\mathrm{OKRidge}}}$ is the of optimal solutions of the **PO** problem (25), and $\boldsymbol{w}_{\phi_{\mathrm{OKRidge}}}$ is the optimal solutions of the **AO** problem (27). According to Theorem 3.2, if $\|\boldsymbol{w}_{\phi_{\mathrm{OKRidge}}}\|_2 \xrightarrow{P} \alpha^*$, we have $\|\boldsymbol{w}_{\Phi_{\mathrm{OKRidge}}}\|_2 \xrightarrow{P} \alpha^*$. Thus, we can analyze the minimizer of **AO** problem (27) instead of **PO** problem (25).

## 5.2 Simplification for AO

In this chapter, we simplify the **AO** problem (27) into ones involving only scalar quantities. Since $\gamma(\boldsymbol{\beta})$ is a convex (see Lemma 5.1), $\partial \gamma(\boldsymbol{\beta}^*)$ is nonempty, convex and compact. According to [13, Theorem 23.4], we have $\gamma(\boldsymbol{\beta}^* + \boldsymbol{w}) = \gamma(\boldsymbol{\beta}^*) + \max_{\boldsymbol{s} \in \partial \gamma(\boldsymbol{\beta}^*)} \boldsymbol{s}^\top \boldsymbol{w} + O(\|\boldsymbol{w}\|_2^2)$. The first-order approximation of $\gamma(\boldsymbol{\beta})$ around the vector of interest $\boldsymbol{\beta}^*$ is

$$\hat{\gamma}(\boldsymbol{\beta}^* + \boldsymbol{w}) := \gamma(\boldsymbol{\beta}^*) + \max_{\boldsymbol{s} \in \partial \gamma(\boldsymbol{\beta}^*)} \boldsymbol{s}^\top \boldsymbol{w}. \tag{28}$$

where $\boldsymbol{\beta} = \boldsymbol{\beta}^* + \boldsymbol{w}$. Then, following the approach from [12], the **AO** problem (27) can be simplified by the first-order approximation (28):

$$\hat{\phi}_{\text{OKRidge}}(\boldsymbol{g}, \boldsymbol{h}) = \min_{\boldsymbol{w}} \max_{\boldsymbol{u}} \frac{1}{\sqrt{n}} \Big[ (\|\boldsymbol{w}\|_2 \boldsymbol{g} - \boldsymbol{\epsilon})^\top \boldsymbol{u} + \|\boldsymbol{u}\|_2 \boldsymbol{h}^\top \boldsymbol{w} + \lambda \big( \gamma(\boldsymbol{\beta}^*) + \max_{\boldsymbol{s} \in \partial \gamma(\boldsymbol{\beta}^*)} \boldsymbol{s}^\top \boldsymbol{w} \big) - \frac{\|\boldsymbol{u}\|_2^2}{4} \Big]$$

$$= \min_{\boldsymbol{w}} \max_{\substack{\|\boldsymbol{u}\|_2 \geq 0 \\ \boldsymbol{s} \in \partial \gamma(\boldsymbol{\beta}^*)}} \frac{1}{\sqrt{n}} \Big[ (\|\boldsymbol{w}\|_2 \boldsymbol{g} - \boldsymbol{\epsilon})^\top \boldsymbol{u} + (\|\boldsymbol{u}\|_2 \boldsymbol{h} + \lambda \boldsymbol{s})^\top \boldsymbol{w} + \lambda \gamma(\boldsymbol{\beta}^*) - \frac{\|\boldsymbol{u}\|_2^2}{4} \Big]. \quad (29)$$

Suppose $f(\boldsymbol{\beta})$ and $\hat{f}(\boldsymbol{\beta})$ denote the objective functions of the original and the approximated **AO** problems (27) and (29), respectively,

$$f(\boldsymbol{\beta}) = (\|\boldsymbol{w}\|_2 \boldsymbol{g} - \boldsymbol{\epsilon})^\top \boldsymbol{u} + \|\boldsymbol{u}\|_2 \boldsymbol{h}^\top \boldsymbol{w} - \frac{\|\boldsymbol{u}\|_2^2}{4} + \lambda \gamma(\boldsymbol{\beta}^* + \boldsymbol{w}),$$

$$\hat{f}(\boldsymbol{\beta}) = (\|\boldsymbol{w}\|_2 \boldsymbol{g} - \boldsymbol{\epsilon})^\top \boldsymbol{u} + \|\boldsymbol{u}\|_2 \boldsymbol{h}^\top \boldsymbol{w} - \frac{\|\boldsymbol{u}\|_2^2}{4} + \lambda \big( \gamma(\boldsymbol{\beta}^*) + \max_{\boldsymbol{s} \in \partial \gamma(\boldsymbol{\beta}^*)} \boldsymbol{s}^\top \boldsymbol{w} \big).$$

Then, based on (28), we have

$$\lim_{\|\boldsymbol{\beta} - \boldsymbol{\beta}^*\|_2 \to 0} \hat{f}(\boldsymbol{\beta}) = f(\boldsymbol{\beta}). \quad (30)$$

Compared with **AO** problem (27), the approximated **AO** problem (29) is tight when $\|\boldsymbol{\beta} - \boldsymbol{\beta}^*\|_2 \to 0$, and we later demonstrate that this condition is satisfied as $\sigma^2 \to 0$, independent of the original **AO** problem (27). This fact allows us to translate the analysis on the optimal solution $\boldsymbol{w}_{\hat{\phi}_{\text{OKRidge}}}$ of the approximated **AO** problem (29) to the analysis on the optimal solution $\boldsymbol{w}_{\phi_{\text{OKRidge}}}$ of the corresponding original **AO** problem (27). Because $\gamma(\boldsymbol{\beta}^*)$ is a constant, the approximated **AO** problem (29) is equivalent to the following optimization problem:

$$\min_{\boldsymbol{w}} \max_{\substack{\|\boldsymbol{u}\|_2 \geq 0 \\ \boldsymbol{s} \in \partial \gamma(\boldsymbol{\beta}^*)}} \frac{1}{\sqrt{n}} \Big[ (\|\boldsymbol{w}\|_2 \boldsymbol{g} - \boldsymbol{\epsilon})^\top \boldsymbol{u} + (\|\boldsymbol{u}\|_2 \boldsymbol{h} + \lambda \boldsymbol{s})^\top \boldsymbol{w} - \frac{\|\boldsymbol{u}\|_2^2}{4} \Big], \quad (31)$$

where we have approximated $\gamma$ in the first order. Since $\boldsymbol{\epsilon} \sim \mathcal{N}(0, \sigma^2 \boldsymbol{I})$, the term $\|\boldsymbol{w}\|_2 \boldsymbol{g} - \boldsymbol{\epsilon}$ above is statistically identical to a random vector with entries drawn i.i.d. from $\mathcal{N}(0, \|\boldsymbol{w}\|_2^2 + \sigma^2)$, where $\boldsymbol{I}$ is the unit matrix. Following the method used by [42], we substitute the first term in the objective (31) with $\sqrt{\|\boldsymbol{w}\|_2^2 + \sigma^2} \boldsymbol{g}^\top \boldsymbol{u}$. Then, we obtain:

$$\min_{\boldsymbol{w}} \max_{\substack{\|\boldsymbol{u}\|_2 \geq 0 \\ \boldsymbol{s} \in \partial \gamma(\boldsymbol{\beta}^*)}} \frac{1}{\sqrt{n}} \Big[ \sqrt{\|\boldsymbol{w}\|_2^2 + \sigma^2} \cdot \boldsymbol{g}^\top \boldsymbol{u} + (\|\boldsymbol{u}\|_2 \boldsymbol{h} + \lambda \boldsymbol{s})^\top \boldsymbol{w} - \frac{\|\boldsymbol{u}\|_2^2}{4} \Big]. \quad (32)$$

Let $\eta = \|\boldsymbol{u}\|_2$. Since $\max_{\|\boldsymbol{u}\|_2 = \eta} \boldsymbol{g}^\top \boldsymbol{u} = \|\boldsymbol{g}\|_2 \cdot \|\boldsymbol{u}\|_2$ and $\boldsymbol{h} \sim \mathcal{N}(0, \boldsymbol{I})$, in term of mathematical expectation, the optimization (32) can be equivalently expressed as:

$$\min_{\boldsymbol{w}} \max_{\substack{\eta \geq 0 \\ \boldsymbol{s} \in \partial \gamma(\boldsymbol{\beta}^*)}} \frac{1}{\sqrt{n}} \Big[ \sqrt{\|\boldsymbol{w}\|_2^2 + \sigma^2} \|\boldsymbol{g}\|_2 \eta - \eta \big( \boldsymbol{h} - \frac{\lambda}{\eta} \boldsymbol{s} \big)^\top \boldsymbol{w} - \frac{\eta^2}{4} \Big]. \quad (33)$$

The objective (33) is strongly convex in $\boldsymbol{w}$ and (jointly) concave in $\eta$, $\boldsymbol{s}$, and the constraint sets are bounded. Therefore, we can reverse the order of min-max in problem (33) based on [13, Corollary 37.3.2]. Let $\alpha = \|\boldsymbol{w}\|_2$. Since $\min_{\|\boldsymbol{w}\|_2 = \alpha} (-\boldsymbol{h} + \frac{\lambda}{\eta} \boldsymbol{s})^\top \boldsymbol{w} = -\alpha \|\boldsymbol{h} - \frac{\lambda}{\eta} \boldsymbol{s}\|_2$, the optimization (33) can be equivalently reformulated as:

$$\max_{\substack{\eta \geq 0 \\ \boldsymbol{s} \in \partial \gamma(\boldsymbol{\beta}^*)}} \min_{\alpha \geq 0} \frac{1}{\sqrt{n}} \Big( \sqrt{\alpha^2 + \sigma^2} \cdot \|\boldsymbol{g}\|_2 \eta - \alpha \eta \|\boldsymbol{h} - \frac{\lambda}{\eta} \boldsymbol{s}\|_2 - \frac{\eta^2}{4} \Big). \quad (34)$$

Next, we further reverse the order of min-max, as the objective (34) exhibits the desired concave-convex structure. Then, we proceed to maximize over $\boldsymbol{s} \in \partial \gamma(\boldsymbol{\beta}^*)$. Since $\min_{\boldsymbol{s} \in \partial \gamma(\boldsymbol{\beta}^*)} \|\boldsymbol{h} - \frac{\lambda}{\eta} \boldsymbol{s}\|_2 = \text{dist}(\boldsymbol{h}, \frac{\lambda}{\eta} \partial \gamma(\boldsymbol{\beta}^*))$, the optimization problem (34) can alternatively be formulated as:

$$\min_{\alpha \geq 0} \max_{\eta \geq 0} \frac{1}{\sqrt{n}} \Big( \sqrt{\alpha^2 + \sigma^2} \cdot \|\boldsymbol{g}\|_2 \eta - \alpha \eta \cdot \text{dist}(\boldsymbol{h}, \frac{\lambda}{\eta} \boldsymbol{s}) - \frac{\eta^2}{4} \Big). \quad (35)$$

Because both the random components $\|\boldsymbol{g}\|_2$ and $\mathrm{dist}(\boldsymbol{h}, \frac{\lambda}{\eta}\boldsymbol{s})$ are Lipschitz, $\|\boldsymbol{g}\|_2$ concentrates around $\sqrt{n}$ and $\mathrm{dist}(\boldsymbol{h}, \frac{\lambda}{\eta}\boldsymbol{s})$ around $\sqrt{D(\frac{\lambda}{\eta})}$ [43, Lemma B.2]. Suppose, as $d \to \infty$, $\frac{D(\tau)}{n} \to \bar{D}(\tau) \in (0,1)$, $\frac{C(\tau)}{n} \to \bar{C}(\tau)$, and $\Gamma(\eta) := \lim_{n\to\infty} \frac{\eta^2}{4\sqrt{n}}$. Then, the optimal minimizer of (35) converges to the optimal minimizer of the following deterministic optimization in probability [44]:

$$\max_{\eta \geq 0} \min_{\alpha \geq 0} \eta\sqrt{\alpha^2 + \sigma^2} - \alpha\eta\sqrt{\bar{D}(\frac{\lambda}{\eta})} - \Gamma(\eta). \tag{36}$$

Here, we complete the simplifications by reducing the **AO** problem (27) to an equivalent optimization (36) that now only involves two scalar variables: $\alpha$ and $\eta$.

## 5.3 Error Analysis

Based on the analysis above, if the optimal solution of optimization (36) is $\alpha = \alpha^*$, we have $\|\boldsymbol{w}_{\hat{\phi}_{\mathrm{OKRidge}}}\|_2 \xrightarrow{P} \alpha^*$ for approximated **AO** problem (29). If $\alpha^*$ further tends to 0, according to formulation (30) and CGMT, $\|\boldsymbol{w}_{\Phi_{\mathrm{OKRidge}}}\|_2 \xrightarrow{P} \alpha^*$ holds for **PO** problem (25). Then, for the estimation error of OKRidge produced by (22), we have $\|\hat{\boldsymbol{\beta}} - \boldsymbol{\beta}^*\|_2 \xrightarrow{P} \alpha^*$. Therefore, it only remains to obtain the optimal value of $\alpha$ in optimization (36) that plays the role of $\|\boldsymbol{w}\|_2$. Following [45], we conclude the estimation error of OKRidge with Theorem 5.2 below.

**Theorem 5.2.** *Suppose $\boldsymbol{\beta}^*$ is the true weight parameter of the problem (1), $\hat{\boldsymbol{\beta}}$ is the optimal solution to the objective function (22) of OKRidge, $\frac{D(\tau)}{n} \to \bar{D}(\tau) \in (0,1)$, $aNSE := \lim_{\sigma^2\to 0} NSE = \lim_{\sigma^2\to 0} \|\hat{\boldsymbol{\beta}} - \boldsymbol{\beta}^*\|_2^2/\sigma^2$. Define $\lambda_{map}$ is the solution of $map(\tau) = 0$ for $\tau > 0$, then, the estimation error of OKRidge is given by the following probability limit:*

$$\lim_{d\to 0} aNSE \xrightarrow{P} \Delta(\hat{\lambda}), \tag{37}$$

*where $\Delta(\hat{\lambda}) = \frac{\bar{D}(\hat{\lambda})}{1 - \bar{D}(\hat{\lambda})}$, and $\hat{\lambda} = \lambda_{map}$.*

The proof of Theorem 5.2 can be seen in Appendix E.

*Remark* 5.3. In the objective (24) concerning estimation error of OKRidge, $\gamma(\boldsymbol{\beta}) = \mathrm{SumTop}_k(\boldsymbol{\beta} \odot \boldsymbol{\beta})$ and the value of $\boldsymbol{\beta}^*$ is assumed to be known. Then, the analysis on $map(\cdot)$ in Section 3.3 reveals that the form of $map(\cdot)$ and the value of $\lambda_{map}$ are fixed. Thus, $\Delta(\hat{\lambda})$ is a function of $\lambda$. In other words, if the regularizer parameter $\lambda$ of OKRidge is fixed, the NSE limit of OKRidge $\Delta(\hat{\lambda})$ is also fixed. Additionally, Theorem 5.2 also indicates that $\lim_{d\to\infty} \lim_{\sigma\to 0} \|\hat{\boldsymbol{\beta}} - \boldsymbol{\beta}^*\|_2 \xrightarrow{P} 0$, which guarantees the effectiveness of $\hat{\boldsymbol{\beta}}$ learned by OKRidge in accurately estimating $\boldsymbol{\beta}^*$. These results substantiate the theoretical reliability of OKRidge and promote its broad application in the real world.

## 6 Numerical Experiments

In this section, we conduct experiments to verify Theorem 5.2. The experiments contain two aspects: (i) When $\lambda$ is fixed, NSE tends to a fixed constant as $\sigma \to 0$. (ii) When $\sigma \to 0$, NSE is determined by the weight $\lambda$ of the regularizer. In other words, NSE is a function of $\lambda$.

In our experiments, $\boldsymbol{\beta}^*$ is randomly generated with $\|\boldsymbol{\beta}^*\|_0 \leq k$. For $i \in \{1, 2, \cdots, n\}$, $\boldsymbol{x}_i$ is drawn i.i.d. from $\mathcal{N}(0, \boldsymbol{I})$, and $\epsilon_i$ is drawn i.i.d. from $\mathcal{N}(0, \sigma^2)$. According to the $k$-sparse linear regression (1), $y_i = \boldsymbol{x}_i^\top \boldsymbol{\beta}^* + \epsilon_i$, we get dataset $(\boldsymbol{x}_i, y_i)$ with $i = 1, 2, \cdots, n$. Then, we apply OKRidge to get the estimator $\hat{\boldsymbol{\beta}}$ and calculate the NSE by $\|\hat{\boldsymbol{\beta}} - \boldsymbol{\beta}^*\|_2^2/\sigma^2$. The NSE is averaged over 10 trials to evaluate the effectiveness of the OKRidge algorithm. In the main paper, we set $\frac{n}{d} = 0.5$, $\frac{k}{d} = 0.1$, $d = 100$. The computer resources are detailed in Appendix F.1. More experiments with various settings about $\frac{n}{d}$ and $\frac{k}{d}$ can be seen in Appendix F.2.

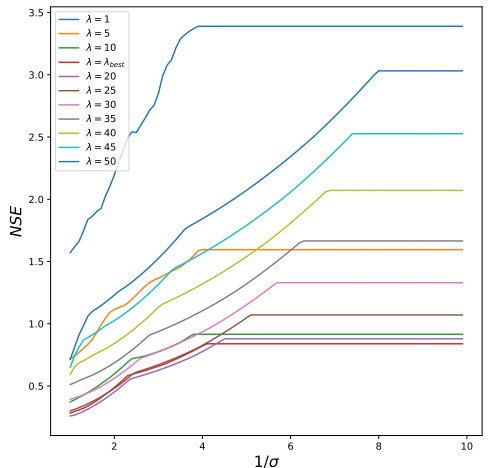 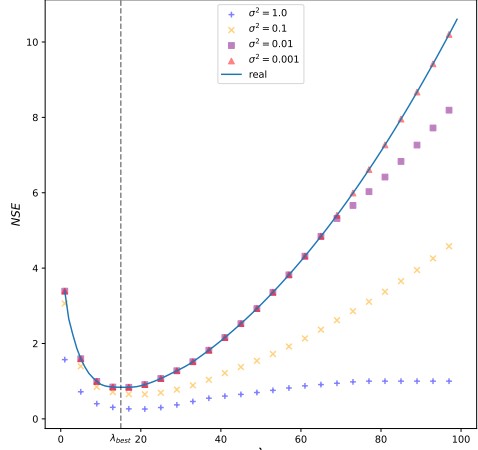

Figure 1: The change of NSE with $1/\sigma$ for OKRidge under different $\lambda$. The red curve at the bottom corresponds to the case $\lambda = \lambda_{best}$.

Figure 2: The change of NSE with $\lambda$ for OKRidge under different $\sigma$. The blue curve corresponds to the real change of $\Delta(\hat{\lambda})$. Here, $\lambda_{best}$ is the optimal weight of the regularizer.

## 6.1 The Change of NSE with $\sigma$

We investigate the change of NSE with $1/\sigma$ under $\lambda = 1, 5, 10, \lambda_{best}, \cdots$. The results are illustrated in Figure 1. As depicted in Figure 1, when $\lambda$ is constant, NSE converges to a fixed value as $\sigma \to 0$. This observation validates aspect (i) of Theorem 5.2.

## 6.2 The Change of NSE with $\lambda$

We analyze the change of NSE with $\lambda$ under $\sigma^2 = 1, 0.1, 0.01, 0.001$. The outcomes are depicted in Figure 2. As shown in Figure 2, the curves converge towards the real blue curve as $\sigma \to 0$, where the blue curve relies on $\lambda$. This observation confirms aspect (ii) of Theorem 5.2.

## 7 Conclusion

In this paper, we present a theoretical high-dimensional error analysis of the OKRidge algorithm in idealized settings using the CGMT framework. Specifically, when OKRidge tackles a $k$-sparse linear model with $\boldsymbol{x} \sim \mathcal{N}(0, \boldsymbol{I})$, $\epsilon \sim \mathcal{N}(0, \sigma^2)$, and $\lim_{d \to \infty} \frac{n}{d} = \delta \in (0, 1)$, we have

$$\lim_{d \to \infty} \lim_{\sigma^2 \to 0} \frac{\|\hat{\boldsymbol{\beta}} - \boldsymbol{\beta}^*\|_2^2}{\sigma^2} \xrightarrow{P} \Delta(\hat{\lambda}), \text{ and } \lim_{d \to \infty} \lim_{\sigma \to 0} \|\hat{\boldsymbol{\beta}} - \boldsymbol{\beta}^*\|_2 \xrightarrow{P} 0.$$

where $\Delta(\hat{\lambda})$ depends on $\lambda$. This indicates that $(i)$ the NSE limit of OKRidge remains constant when $\lambda$ is fixed; $(ii)$ $\hat{\boldsymbol{\beta}}$ learned by OKRidge is effective in estimating $\boldsymbol{\beta}^*$. Our experimental findings support these theoretical assertions. This theoretical error analysis substantiates the reliability of OKRidge and provides guidelines on the error analysis of other algorithms.

## Acknowledgments and Disclosure of Funding

This work is supported by the Key R&D Program of Hubei Province under Grant 2024BAB038, National Key R&D Program of China under Grant 2023YFC3604702, and the Fundamental Research Fund Program of LIESMARS.

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

# The Reliability of OKRidge (Appendix)

## A  Limitaion and Impact

**Limitaion**: Our results rely on Gaussian input settings. For non-Gaussian settings, we can utilize Fisher transformation, Box-Cox transformation, or inversion sampling to transform non-Gaussian distribution to Gaussian distribution. In our future work, we will discuss the potential extensions of our findings to non-Gaussian input settings, providing insights into the universality of the results.

**Impact**: Our work provides theoretical support for the broad application of OKRidge, which does not require proprietary software or expensive licenses [1], unlike its main competitor. This can significantly impact various regression applications.

## B  Proof of Theorem 4.1

*Theorem* 4.1. If we define the parameter $\boldsymbol{c}$ as (16), the problem (9) is equivalent to the following saddle point optimization problem:

$$\min_{\boldsymbol{\beta},\boldsymbol{z}} \mathcal{L}_{\text{ridge}}^{\text{saddle}}(\boldsymbol{\beta},\boldsymbol{z}) \quad \text{s.t.} \quad \sum_{j=1}^{d} z_j \leq k, z_j \in [0,1], \tag{38}$$

where

$$\mathcal{L}_{\text{ridge}}^{\text{saddle}}(\boldsymbol{\beta},\boldsymbol{z}) := \|\boldsymbol{y} - \boldsymbol{X}\boldsymbol{\beta}\|_2^2 + \lambda \sum_{j=1, z_j \neq 0}^{d} \frac{\beta_j^2}{z_j}. \tag{39}$$

*Proof.* The problem (9) is

$$\min_{\boldsymbol{\beta},\boldsymbol{z}} \max_{\boldsymbol{c}} \mathcal{L}_{\text{ridge}}^{\text{Fenchel}}(\boldsymbol{\beta},\boldsymbol{z},\boldsymbol{c}),$$

$$\text{s.t.} \sum_{j=1}^{d} z_j \leq k, \ z_j \in [0,1],$$

where

$$\mathcal{L}_{\text{ridge}}^{\text{Fenchel}}(\boldsymbol{\beta},\boldsymbol{z},\boldsymbol{c}) := \|\boldsymbol{y} - \boldsymbol{X}\boldsymbol{\beta}\|_2^2 + \lambda \sum_{j=1}^{d} (\beta_j c_j - \frac{c_j^2}{4} z_j).$$

The parameter $\boldsymbol{c}$ in (16) is

$$c_j = \begin{cases} \rho \in \mathbb{R} & \text{, if } z_j = 0, \\ \frac{2\beta_j}{z_j} & \text{, if } z_j \neq 0. \end{cases}$$

Substitute parameter $\boldsymbol{c}$ of (16) into (9),

$$\mathcal{L}_{\text{ridge}}^{\text{Fenchel}}(\boldsymbol{\beta},\boldsymbol{z},\boldsymbol{c}) = \|\boldsymbol{y} - \boldsymbol{X}\boldsymbol{\beta}\|_2^2 + \lambda \sum_{j=1}^{d} (\beta_j c_j - \frac{c_j^2}{4} z_j)$$

$$= \|\boldsymbol{y} - \boldsymbol{X}\boldsymbol{\beta}\|_2^2 + \lambda \sum_{j=1, z_j \neq 0}^{d} \left[ \beta_j \cdot \frac{2\beta_j}{z_j} - (\frac{2\beta_j}{z_j})^2 / 4 \cdot z_j \right]$$

$$= \|\boldsymbol{y} - \boldsymbol{X}\boldsymbol{\beta}\|_2^2 + \lambda \sum_{j=1, z_j \neq 0}^{d} \left[ \frac{2\beta_j^2}{z_j} - \frac{\beta_j^2}{z_j} \right]$$

$$= \|\boldsymbol{y} - \boldsymbol{X}\boldsymbol{\beta}\|_2^2 + \lambda \sum_{j=1, z_j \neq 0}^{d} \frac{2\beta_j^2}{z_j}$$

Define
$$\mathcal{L}_{\text{ridge}}^{\text{saddle}}(\boldsymbol{\beta}, \boldsymbol{z}) := \|\boldsymbol{y} - \boldsymbol{X}\boldsymbol{\beta}\|_2^2 + \lambda \sum_{j=1, z_j \neq 0}^{d} \frac{\beta_j^2}{z_j}.$$

Then, the problem (9) can be equivalently written as

$$\min_{\boldsymbol{\beta}, \boldsymbol{z}} \mathcal{L}_{\text{ridge}}^{\text{saddle}}(\boldsymbol{\beta}, \boldsymbol{z}) \quad \text{s.t.} \quad \sum_{j=1}^{d} z_j \leq k, z_j \in [0, 1],$$

which is our saddle point optimization (38). □

## C  Proof of Theorem 4.2

*Theorem* 4.2. The function $\mathcal{L}(\boldsymbol{\beta})$ defined in Equation (20) is lower bounded by
$$\mathcal{L}(\boldsymbol{\beta}) \geq \|\boldsymbol{y} - \boldsymbol{X}\boldsymbol{\beta}\|_2^2 + \lambda \text{SumTop}_k(\boldsymbol{\beta} \odot \boldsymbol{\beta}). \tag{40}$$
where $\odot$ is Hadamard product, and $\text{SumTop}_k(\cdot)$ denotes the summation of the largest $k$ elements of a given vector.

*Proof.* The Equation (20) is

$$\mathcal{L}(\boldsymbol{\beta}) = \min_{\boldsymbol{z}} \mathcal{L}_{\text{ridge}}^{\text{saddle}}(\boldsymbol{\beta}, \boldsymbol{z}),$$

$$\text{s.t.} \sum_{j=1}^{d} z_j \leq k, \ z_j \in [0, 1].$$

According to (20) and (39),

$$\mathcal{L}(\boldsymbol{\beta}) = \min_{\boldsymbol{z}} \mathcal{L}_{\text{ridge}}^{\text{saddle}}(\boldsymbol{\beta}, \boldsymbol{z}) = \min_{\boldsymbol{z}} \|\boldsymbol{y} - \boldsymbol{X}\boldsymbol{\beta}\|_2^2 + \lambda \sum_{j=1, z_j \neq 0}^{d} \frac{\beta_j^2}{z_j} \tag{41}$$

Following the method of [1], if we minimize $z$ in the optimization (41) under the constraints $\sum_{j=1}^{p} z_j \leq k$ and $z_j \in [0, 1]$ for $\forall j$, we have $z_j = 1$ for the top $k$ terms of $\beta_j^2$ and $z_j = 0$ otherwise. Then, we have

$$\mathcal{L}(\boldsymbol{\beta}) \geq \|\boldsymbol{y} - \boldsymbol{X}\boldsymbol{\beta}\|_2^2 + \lambda \sum_{j=1, z_j \neq 0}^{d} \frac{\beta_j^2}{1} = \|\boldsymbol{y} - \boldsymbol{X}\boldsymbol{\beta}\|_2^2 + \lambda \text{SumTop}_k(\boldsymbol{\beta} \odot \boldsymbol{\beta}).$$

□

## D  Proof of Lemma 5.1

*Lemma* 5.1. Suppose $\psi(\boldsymbol{w}, \boldsymbol{u})$ is defined as formulation (26). Then, $\psi(\boldsymbol{w}, \boldsymbol{u})$ is convex-concave on $\mathbb{R}^d \times \mathbb{R}^n$.

*Proof.* Obviously, $\psi(\boldsymbol{w}, \boldsymbol{u})$ is concave about $\boldsymbol{u}$. Next, we indicate that $\psi(\boldsymbol{w}, \boldsymbol{u})$ is convex about $\boldsymbol{w}$. For any $\bar{\boldsymbol{w}}, \widetilde{\boldsymbol{w}} \in \mathbb{R}^d$ and $\forall \theta \in (0, 1)$,
$$[\theta \bar{w}_j + (1 - \theta)\widetilde{w}_j]^2 \leq \theta \bar{w}_j^2 + (1 - \theta)\widetilde{w}_j^2.$$
Then, we have

$$\text{SumTop}_k\big([\theta\bar{\boldsymbol{w}} + (1 - \theta)\widetilde{\boldsymbol{w}}] \odot [\theta\bar{\boldsymbol{w}} + (1 - \theta)\widetilde{\boldsymbol{w}}]\big)$$
$$\leq \text{SumTop}_k\big(\theta\bar{\boldsymbol{w}} \odot \bar{\boldsymbol{w}} + (1 - \theta)\widetilde{\boldsymbol{w}} \odot \widetilde{\boldsymbol{w}}\big)$$
$$\leq \text{SumTop}_k\big(\theta\bar{\boldsymbol{w}} \odot \bar{\boldsymbol{w}}\big) + \text{SumTop}_k\big((1 - \theta)\widetilde{\boldsymbol{w}} \odot \widetilde{\boldsymbol{w}}\big)$$
$$\leq \theta\text{SumTop}_k\big(\bar{\boldsymbol{w}} \odot \bar{\boldsymbol{w}}\big) + (1 - \theta)\text{SumTop}_k\big(\widetilde{\boldsymbol{w}} \odot \widetilde{\boldsymbol{w}}\big).$$

$\text{SumTop}_k(\cdot)$ is a convex operator. Therefore, $\psi(\boldsymbol{w}, \boldsymbol{u})$ is concave about $\boldsymbol{u}$ and convex about $\boldsymbol{w}$.

□

# E Proof of Theorem 5.2

*Theorem* 5.2. Suppose $\boldsymbol{\beta}^*$ is the true weight parameter of the problem (1), $\hat{\boldsymbol{\beta}}$ is the optimal solution to the objective function (22) of OKRidge, $\frac{D(\tau)}{n} \to \bar{D}(\tau) \in (0,1)$,

$$\text{aNSE} := \lim_{\sigma^2 \to 0} \text{NSE} = \lim_{\sigma^2 \to 0} \|\hat{\boldsymbol{\beta}} - \boldsymbol{\beta}^*\|_2^2 / \sigma^2.$$

Define $\lambda_{map}$ is the solution of $\text{map}(\tau) = 0$ for $\tau > 0$, then, the estimation error of OKRidge is given by the following probability limit:

$$\lim_{d \to 0} \text{aNSE} \xrightarrow{P} \Delta(\hat{\lambda}), \tag{42}$$

where $\Delta(\hat{\lambda}) = \frac{\bar{D}(\hat{\lambda})}{1 - \bar{D}(\hat{\lambda})}$, and $\hat{\lambda} = \lambda_{map}$.

*Proof.* Starting from the simplified **AO** problem (36), let's define

$$\kappa(\alpha, \eta) := \eta\sqrt{\alpha^2 + \sigma^2} - \alpha\eta\sqrt{\bar{D}(\frac{\lambda}{\eta})} - \Gamma(\eta). \tag{43}$$

Given that $\kappa(\alpha, \eta)$ is strongly convex in $\alpha$ and concave in $\eta$, we denote $(\alpha^*, \eta^*)$ as the Nash equilibrium of $\kappa(\alpha, \eta)$. Then, $\alpha^*$ is unique and we use duality to compute $\alpha^*$. For any $\eta$,

$$\frac{\partial \kappa(\alpha, \eta)}{\partial \alpha} = \frac{\alpha\eta}{\sqrt{\alpha^2 + \sigma^2}} - \eta\sqrt{\bar{D}(\frac{\lambda}{\eta})} = 0, \Rightarrow \alpha^*(\eta) = \sigma\sqrt{\frac{\bar{D}(\frac{\lambda}{\eta})}{1 - \bar{D}(\frac{\lambda}{\eta})}}, \tag{44}$$

which is well-defined, due to $1 - \bar{D}(\frac{\lambda}{\eta}) > 0$. Substituting $\alpha^*(\eta)$ back into (43) gives:

$$\kappa(\alpha^*, \eta) = \eta\sqrt{\alpha^{*2} + \sigma^2} - \alpha^*\eta\sqrt{\bar{D}(\frac{\lambda}{\eta})} - \Gamma(\eta)$$

$$= \eta\sqrt{\sigma^2 \frac{\bar{D}(\frac{\lambda}{\eta})}{1 - \bar{D}(\frac{\lambda}{\eta})} + \sigma^2} - \sigma\eta\sqrt{\frac{\bar{D}(\frac{\lambda}{\eta})}{1 - \bar{D}(\frac{\lambda}{\eta})}} \cdot \sqrt{\bar{D}(\frac{\lambda}{\eta})} - \Gamma(\eta)$$

$$= \frac{\eta\sigma}{\sqrt{1 - \bar{D}(\frac{\lambda}{\eta})}} - \frac{\eta\sigma\bar{D}(\frac{\lambda}{\eta})}{\sqrt{1 - \bar{D}(\frac{\lambda}{\eta})}} - \Gamma(\eta) = \eta\sigma\sqrt{1 - \bar{D}(\frac{\lambda}{\eta})} - \Gamma(\eta).$$

Differentiating $\kappa(\alpha^*, \eta)$ with respect to $\eta$, we have

$$\frac{\partial \kappa(\alpha^*, \eta)}{\partial \eta} = \sigma\sqrt{1 - \bar{D}(\frac{\lambda}{\eta})} + \frac{1}{2}\eta\sigma\frac{C(\frac{\lambda}{\eta})}{\sqrt{1 - \bar{D}(\frac{\lambda}{\eta})}} \cdot \frac{\lambda}{\eta^2} - \Gamma'(\eta) = \sigma\frac{1 - \bar{C}(\frac{\lambda}{\eta}) - \bar{D}(\frac{\lambda}{\eta})}{\sqrt{1 - \bar{D}(\frac{\lambda}{\eta})}} - \Gamma'(\eta)$$

$$= \sigma\text{map}(\frac{\lambda}{\eta}) - \lim_{n \to \infty} \frac{\eta}{2\sqrt{n}}. \tag{45}$$

Here, $\text{map}(\tau) = 0$ when $\tau = \lambda_{map}$. Hence, $\eta^* = \frac{\lambda}{\lambda_{map}}$ is the optimal solution, due to

$$\frac{\partial \kappa(\alpha^*, \eta)}{\partial \eta}\Big|_{\eta=\eta^*} = \sigma\text{map}(\lambda_{map}) - \lim_{n \to \infty} \frac{\lambda}{2\lambda_{map}\sqrt{n}} = 0.$$

Because the form of $\gamma(\cdot)$ and the value of $\boldsymbol{\beta}^*$ are determined, the form of $map(\cdot)$ and the value of $\lambda_{map}$ are fixed. In other words, $\lambda_{map}$ is a function about $\lambda$. Therefore, we can take $\eta^* = \frac{\lambda}{\lambda_{map}}$ to formulation (44),

$$\alpha^*(\eta^*) = \sigma\sqrt{\frac{\bar{D}(\frac{\lambda}{\eta^*})}{1 - \bar{D}(\frac{\lambda}{\eta^*})}} = \sigma\sqrt{\frac{\bar{D}(\lambda_{map})}{1 - \bar{D}(\lambda_{map})}}.$$

Moreover $\kappa(\alpha^*, \eta^*) = \frac{\lambda}{\lambda_{map}} \sigma \sqrt{1 - \bar{D}(\lambda_{map})}$. Denote $\hat{\lambda} = \lambda_{map}$,

$$\alpha^* = \sigma \sqrt{\frac{\bar{D}(\hat{\lambda})}{1 - \bar{D}(\hat{\lambda})}}. \tag{46}$$

Based on the analysis above, if the optimal solution of optimization (36) is $\alpha = \alpha^*$, we have $\|\boldsymbol{w}_{\hat{\phi}_{\text{OKRidge}}}\|_2 \xrightarrow{P} \alpha^*$ for approximated **AO** problem (29). Furthermore, according to formulation (46), $\lim_{\sigma \to 0} \alpha^* = 0$ occurs for the approximated **AO** problem (29) and is independent of the original **AO** problem (27). Thus, formulation (30) holds:

$$\lim_{\|\boldsymbol{\beta} - \boldsymbol{\beta}^*\|_2 \to 0} \hat{f}(\boldsymbol{\beta}) = f(\boldsymbol{\beta}) \Leftrightarrow \lim_{d \to \infty} \lim_{\sigma \to 0} \hat{f}(\boldsymbol{\beta}) = f(\boldsymbol{\beta}). \tag{47}$$

In the case $n \to \infty$ and $\sigma \to 0$, formulation (47) allows us to translate the optimal error $\alpha^*$ of the approximated **AO** problem (29) to the optimal error of the original **AO** problem (27). Combining formulations (46), (47), and CGMT, $\|\boldsymbol{w}_{\Phi_{\text{OKRidge}}}\|_2 \xrightarrow{P} \alpha^*$ holds for **PO** problem (25). Then, for the estimation error of OKRidge produced by (22), we have

$$\|\hat{\boldsymbol{\beta}} - \boldsymbol{\beta}^*\|_2 \xrightarrow{P} \alpha^*.$$

Therefore, combining **PO** problem (25), **AO** problem (27), the relation (47), and the formulations (23) and (46), according to CGMT, we obtain::

$$\lim_{d \to \infty} \lim_{\sigma \to 0} \|\hat{\boldsymbol{\beta}} - \boldsymbol{\beta}^*\|_2 / \sigma \xrightarrow{P} \sqrt{\frac{\bar{D}(\hat{\lambda})}{1 - \bar{D}(\hat{\lambda})}} \Rightarrow \lim_{d \to \infty} \lim_{\sigma \to 0} \|\hat{\boldsymbol{\beta}} - \boldsymbol{\beta}^*\|_2^2 / \sigma^2 \xrightarrow{P} \frac{\bar{D}(\hat{\lambda})}{1 - \bar{D}(\hat{\lambda})}.$$

To sum up,

$$\lim_{d \to 0} \text{aNSE} \xrightarrow{P} \Delta(\hat{\lambda}),$$

where $\Delta(\hat{\lambda}) = \frac{\bar{D}(\hat{\lambda})}{1 - \bar{D}(\hat{\lambda})}$, and $\hat{\lambda} = \lambda_{map}$.

$\square$

# F  Experiments Appendix

## F.1  Computing Platform

All experiments were run on the 10x TensorEX TS2-673917-DPN Intel Xeon Gold 6226 Processor, 2.7Ghz. We set the memory limit to be 100GB.

## F.2  More Experiments

Figures F1 $\sim$ F7 (a) demonstrate that when $\lambda$ is fixed, NSE converges to a constant as $\sigma \to 0$. In Figures F1 $\sim$ F7 (b), the curves converge towards the real blue curve as $\sigma \to 0$, with the real blue curve representing $\Delta(\hat{\lambda})$. These observation validate Theorem 5.2.

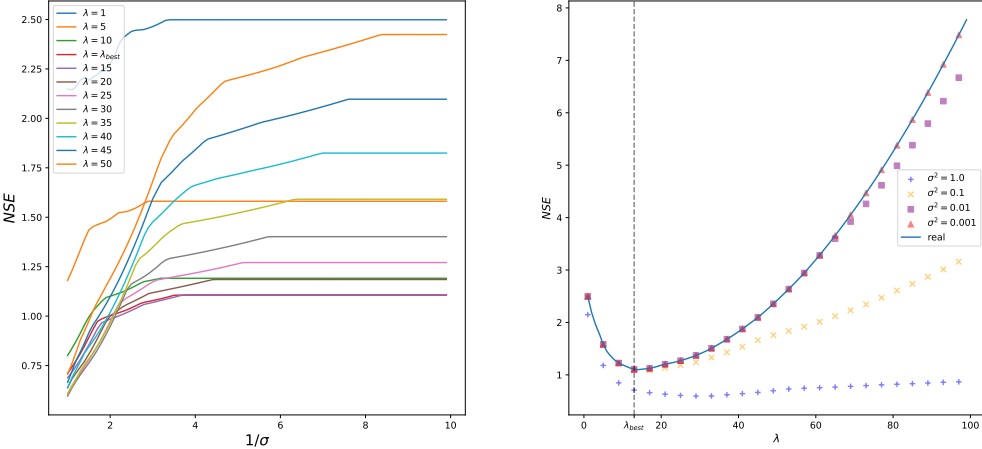

(a) The change of NSE with $1/\sigma$ under different $\lambda$.

(b) The change of NSE with $\lambda$ under different $\sigma$.

Figure F1: The change of NSE under $\frac{n}{d} = 0.5$, $\frac{k}{d} = 0.1$, $d = 200$

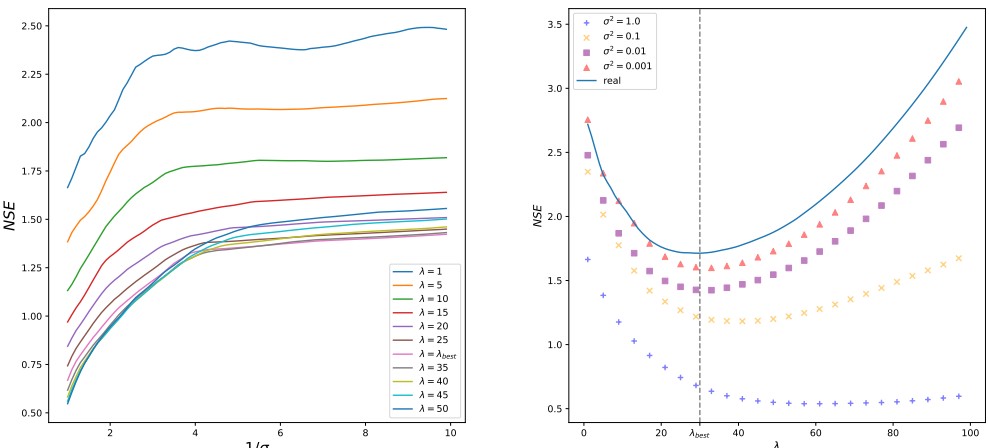

(a) The change of NSE with $1/\sigma$ under different $\lambda$.

(b) The change of NSE with $\lambda$ under different $\sigma$.

Figure F2: The change of NSE under $\frac{n}{d} = 0.5$, $\frac{k}{d} = 0.1$, $d = 1000$

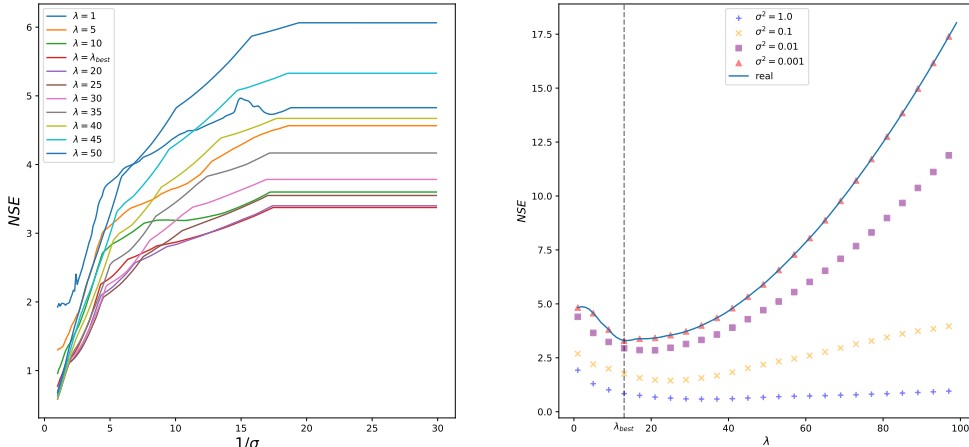

(a) The change of NSE with $1/\sigma$ under different $\lambda$.

(b) The change of NSE with $\lambda$ under different $\sigma$.

Figure F3: The change of NSE under $\frac{n}{d} = 0.4$, $\frac{k}{d} = 0.1$, $d = 1000$

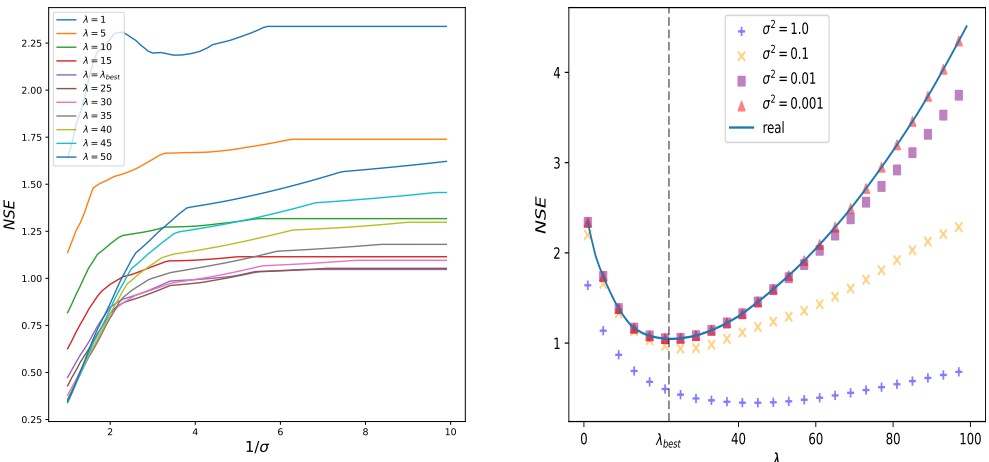

(a) The change of NSE with $1/\sigma$ under different $\lambda$.

(b) The change of NSE with $\lambda$ under different $\sigma$.

Figure F4: The change of NSE under $\frac{n}{d} = 0.6$, $\frac{k}{d} = 0.1$, $d = 1000$

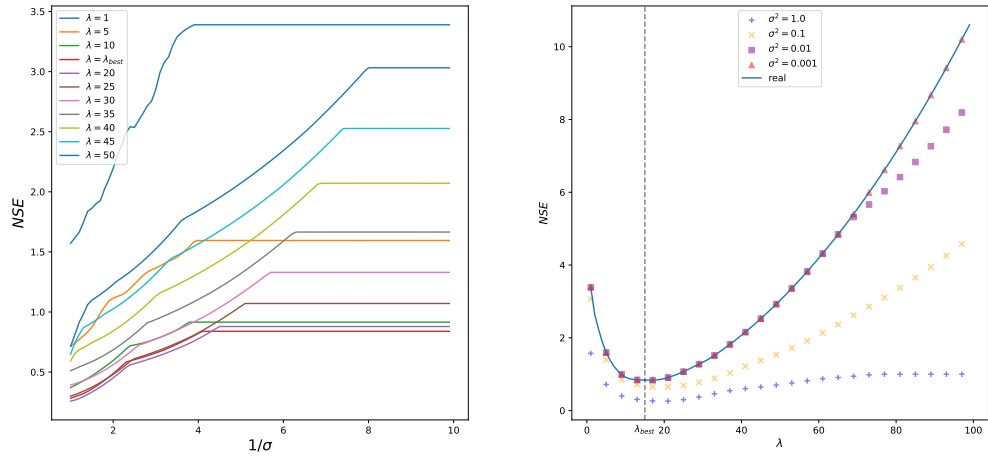

(a) The change of NSE with $1/\sigma$ under different $\lambda$.

(b) The change of NSE with $\lambda$ under different $\sigma$.

Figure F5: The change of NSE under the sparsity $\frac{k}{d} = 0.10$

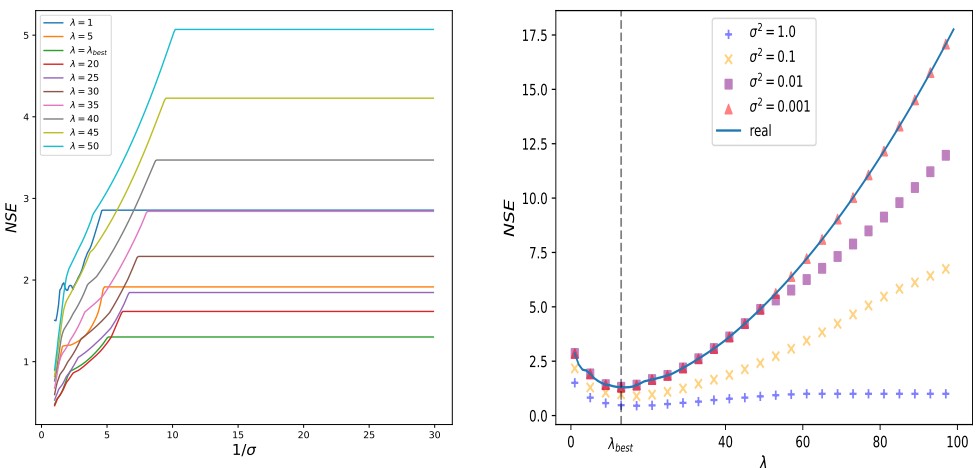

(a) The change of NSE with $1/\sigma$ under different $\lambda$.

(b) The change of NSE with $\lambda$ under different $\sigma$.

Figure F6: The change of NSE under the sparsity $\frac{k}{d} = 0.15$

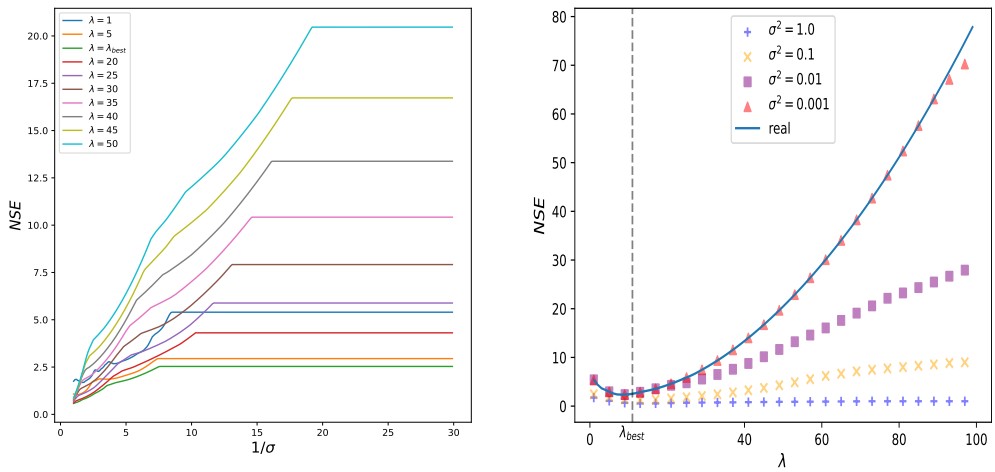

(a) The change of NSE with $1/\sigma$ under different $\lambda$.

(b) The change of NSE with $\lambda$ under different $\sigma$.

Figure F7: The change of NSE under the sparsity $\frac{k}{d} = 0.20$

