# OpenReview forum: "The Reliability of OKRidge Method in Solving Sparse Ridge Regression Problems"
_NeurIPS.cc/2024/Conference — NeurIPS 2024 poster_

### Official Review · Reviewer_8imU · 2024-07-03

**Soundness:** 3
**Presentation:** 3
**Contribution:** 4
**Rating:** 8
**Confidence:** 5

**Summary:**

This paper analyzes the estimation error of the Scalable Optimal K-Sparse Ridge Regression (OKRidge) method proposed by [1]. Specifically, they reframe the estimation error of OKRidge as a Primary Optimization (PO) problem and use the Convex Gaussian Min-Max Theorem (CGMT) to simplify the PO problem into an Auxiliary Optimization (AO) problem. Subsequently, this paper provides a theoretical error analysis for OKRidge based on the AO problem.

**Strengths:**

__Originality:__ The absence of theoretical analysis on the error of OKRidge impedes its large-scale applications. This paper analyzes the estimation error of OKRidge to fill this gap.

__Clarity:__ This paper demonstrates a high level of clarity in its presentation, which significantly enhances its readability. All symbols are clearly defined at the outset and the proofs provided are rigorous and detailed.

__Quality and Significance:__ This paper applies CGMT technology to transform the estimation error of OKRidge into a simplified AO problem and offer a theoretical error analysis for OKRidge. This theoretical error analysis substantiates the reliability of OKRidge and provides guidelines for the error analysis of other algorithms.

**Weaknesses:**

1. Many formulas lack punctuation marks after them, such as formulations (10), (11), (22), etc. The author should review the entire text and pay attention to these details.

2. Theorem 5.2 demonstrates that $\hat{\lambda}=\lambda_{map}$. It would be more concise to substitute $\Delta(\hat{\lambda})$ with $\Delta(\lambda_{map})$.

3. The colors of the curves for $\lambda=1$ and $\lambda=50$ in Figure 1 and F1-F4 (a) are the same. This paper should ensure that the curves are distinguishable by using different colors.

**Questions:**

The theoretical results are based on CGMT technology, which requires the input to be Gaussian. Could you generalize the input to non-Gaussian distributions?

**Limitations:**

Yes, the authors have adequately addressed the limitations.

---

> ### Author Rebuttal · Authors · 2024-08-05
>
> ## Answer to Reviewer 8imU
>
> Dear Reviewer 8imU,
>
> Thank you for your job in reviewing our paper. We are very sorry for the inconvenience caused by our presentations. We extend our heartfelt gratitude for your patience and meticulous guidance. Your insightful comments is valuable for us and we appreciate the opportunity to address your questions and concerns.
>
> ### In regards to your Weaknesses:
>
>
> __Weakness 1.__  Many formulas lack punctuation marks after them, such as formulations (10), (11), (22), etc. The author should review the entire text and pay attention to these details.
>
> __Answer:__  Thank you for your comments. We have checked the entire paper and rectified these details in our revision.
>
> __Weakness 2.__ Theorem 5.2 demonstrates that $\hat{\lambda}=\lambda_{map}$. It would be more concise to substitute $\Delta(\hat{\lambda})$ with $\Delta(\lambda_{map})$.
>
> __Answer:__ Thank you for your suggestion. Following your suggestion, we will replace $\Delta (\lambda)$ with $\Delta (\hat\lambda)$ in our revision to make it more clear.
>
> __Weakness 3.__ The colors of the curves for $\lambda=1$ and $\lambda=50$ in Figure 1 and F1-F4 (a) are the same. This paper should ensure that the curves are distinguishable by using different colors.
>
> __Answer:__ Thank you for your suggestion. We will rectify this to ensure that the curves are distinguishable by using different colors in our revision.
>
>
> ### In regards to your Questions:
>
> __Question 1.__ The theoretical results are based on CGMT technology, which requires the input to be Gaussian. Could you generalize the input to non-Gaussian distributions?
>
> __Answer:__ Thank you for your comments. The OKRidge proposed by [1] is currently the state-of-the-art algorithm for solving sparse ridge regression problems. However, [1] lacks theoretical analysis on the error of the OKRidge method. To the best of our knowledge, we are the first to investigate the estimation error of the OKRidge method. The primary contribution of our paper is to provide a theoretical perspective on the error of the OKRidge method under Gaussian assumption. Extending to general conditions like non-Gaussian distributions is beyond the scope of this paper.
>
> For non-Gaussian settings, we can utilize Fisher transformation, Box-Cox transformation, or inversion sampling to transform non-Gaussian distribution to Gaussian distribution. We will generalize the input to non-Gaussian distributions in our future work.

---

> > ### Comment · Area_Chair_fmWz · 2024-08-12
> >
> > Reviewer 8imU:
> >
> > Can you please respond to the rebuttal as soon as possible? Your comments will be greatly appreciated. Many thanks,
> >
> > AC

---

> > ### Comment · Reviewer_8imU · 2024-08-13
> >
> > Thanks for the author's reply, my concerns have been well answered. Thanks.

---

> > > ### Author Response · Authors · 2024-08-13
> > >
> > > Thank you for your acknowledgment and effort.

---

### Official Review · Reviewer_mBVh · 2024-07-06

**Soundness:** 2
**Presentation:** 1
**Contribution:** 3
**Rating:** 3
**Confidence:** 2

**Summary:**

An error analysis of a lower bound technique for solving sparse ridge regression problems is presented. Sparse ridge regression is ridge regression with the constraint that the parameter vector has at most k non-zero entries, where k is a parameter. It has been proposed to solve the sparse ridge regression problem by solving a "tight lower" bound problem. The normalized deviation of the solution of the lower bound problem to the "true" model parameter vector is analyzed in the high-dimensional setting, that is, the setting where the number of features (and data points) go to infinity.

**Strengths:**

The problem that is addressed by the paper is interesting (for an expert community) and technically challenging.

**Weaknesses:**

I am not able to thoroughly review the paper in the time available for a NeurIPS review. However, since the contributions of the paper are mostly theoretical, a careful review by someone closer to the topic would be necessary to make sure that the paper is sound. Probably a journal is a better venue for this type paper.  Nonetheless, I would have tried to check technical details of the paper if it had been more accessible. The presentation is fairly poor, which makes it difficult (and time-consuming) for a non-expert like me to thoroughly review it.

Some areas where the presentation can be improved:
- Abstracts should not include references.
- The introduction immediately becomes technical, which makes it difficult to understand that problem that you are addressing. Make clear that you provide a theoretical high-dimensional analysis of an algorithm in an idealized setting. You are analyzing the case, where the entries of data matrix are chosen i.i.d from standard normal distributions. Another assumption is that the number n(d) of data points grows with d such that n(d)/d = \delta \in (0,1), where \delta is a constant. All this information is there in the paper, but it could be put better into perspective if your goals had been clearly stated at this point.
- Make clear that the estimation error in Equation 5 is a random variable, because X and \epsilon are random.
- Equation 5 is missing a \odot
- The main paper provides many technical definitions/details but no proofs. The details are relevant when checking the proofs, but do not help with the intuition.

**Questions:**

1. When you solve Problem 4 with the objective function from Equation 3, do you set the n-k smallest entries in \hat\beta to zero?
2. In Line 56, why is it \Delta (\hat\lambda) and not \Delta (\lambda)?

**Limitations:**

None. This is a theory paper.

---

> ### Author Rebuttal · Authors · 2024-08-05
>
> ## Answer to Reviewer mBVh
>
> Dear Reviewer mBVh,
>
> We truly appreciate the patience and effort you've dedicated to providing valuable feedback. Your meticulous guidance is greatly valuable for us to enhance the overall quality of our research. We appreciate the opportunity to address your questions and concerns.
>
> ### In regards to your Weaknesses proposed to improve presentation:
>
> __Weakness 1.__ Abstracts should not include references.
>
> __Answer:__ Thank you for your comments. Following your suggestion, we will delete the references in the abstraction in our revision.
>
> __Weakness 2.__ The introduction immediately becomes technical, which makes it difficult to understand the problem that you are addressing. Make clear that you provide a theoretical high-dimensional analysis of an algorithm in an idealized setting. You are analyzing the case, where the entries of data matrix are chosen i.i.d from standard normal distributions. Another assumption is that the number n(d) of data points grows with d such that $n(d)/d = \delta \in (0,1)$, where $\delta$ is a constant. All this information is there in the paper, but it could be put better into perspective if your goals had been clearly stated at this point.
>
> __Answer:__ Thank you for your comments. We apologize for any inconvenience caused by our presentations. Following your suggestion, we make our goals more clearly stated: We provide a theoretical high-dimensional analysis of the OKRidge algorithm in idealized settings that the entries of the data matrix are chosen i.i.d from standard normal distributions and $\lim_{d\to\infty}n(d)/d = \delta \in (0,1)$. We will emphasize this point in our revision.
>
>
> __Weakness 3.__ Make clear that the estimation error in Equation 5 is a random variable, because $X$ and $\epsilon$ are random.
>
> __Answer:__ Thank you for your suggestion. In our original paper, the estimation error in Equation (5) defaults to a random variable. We are sorry for the inconvenience caused by our presentations. Following your suggestion, in our revised version, we will emphasize that the estimation error $w$ in Equation (5) is a random variable with randomness from the random variables $X$ and $\epsilon$.
>
>
> __Weakness 4.__ Equation 5 is missing a ``$\odot$''.
>
> __Answer:__ Thank you for your comments. We will add $\odot$ to rectify Equation (5) in our revision.
>
>
> __Weakness 5.__ The main paper provides many technical definitions/details but no proofs. The details are relevant when checking the proofs, but do not help with the intuition.
>
> __Answer:__ Thank you for your insightful feedback on our manuscript. Our paper is purely theoretical, with technical details and proofs provided in our Appendix B~E (See lines 398-450). We have also marked the locations of the proofs for the key steps in the original paper. We apologize for any inconvenience caused by our presentations. Following your suggestions, we will provide more intuitions for our technical details in our revision.
>
> ### In regards to your Questions:
>
> __Question 1.__ When you solve Problem 4 with the objective function from Equation 3, do you set the $n-k$ smallest entries in $\hat\beta$ to zero?
>
> __Answer:__ Thank you for your comments. According to our Theorem 5.2, $\lim_{d\to \infty}\lim_{\sigma\to 0}\Vert\hat{\pmb{\beta}}-\pmb{\beta}^*\Vert_2\stackrel{P}{\longrightarrow} 0$. Therefore, when $n(d)$ is sufficiently large, the $n-k$ smallest entries in $\hat\beta$ will become zero. We do not manually set the $n-k$ smallest entries in $\hat\beta$ to zero.
>
> __Question 2.__ In Line 56, why is it $\Delta (\hat\lambda)$ and not $\Delta (\lambda)$?
>
> __Answer:__  Thank you for your comments. According to our Theorem 5.2, we have $\hat\lambda=\lambda_{map}$. Although $\lambda_{map}$ is decided by $\lambda$, they have different meanings. Therefore, it is $\Delta (\hat\lambda)$, not $\Delta (\lambda)$. We will emphasize this in our revision.
>
> We sincerely thank you once again for your time and effort in reviewing our paper. The opportunity to solve your concerns undoubtedly contributes to my personal growth. We hope that our answers have met your expectations and satisfaction.

---

> > ### Author Response · Authors · 2024-08-09
> > **Prompt response**
> >
> > Dear Reviewer mBVh,
> >
> > We hope this message finds you well. I am reaching out to kindly request your prompt response to confirm whether our responses adequately address your queries. We sincerely thank you for your time and effort during this discussion period. Your timely feedback is greatly appreciated.

---

> > > ### Comment · Reviewer_mBVh · 2024-08-09
> > >
> > > It has been a while since I read your paper. I will check it again and come back to you tomorrow.

---

> > > > ### Author Response · Authors · 2024-08-10
> > > > **Appreciation for Your Continued Review and Feedback**
> > > >
> > > > Dear Reviewer mBVh,
> > > >
> > > > Thank you very much for taking the time to review my paper and for your thoughtful consideration. I understand that it may require additional effort on your part, and I truly appreciate your dedication to providing feedback. We hope that our responses have met your expectations and addressed your concerns. Please do not hesitate to reach out if you have any further questions or need any additional information from me. I am looking forward to discussing with you.
> > > >
> > > > I apologize for any inconvenience caused by rechecking our paper, and I sincerely thank you once again for your time and effort.
> > > >
> > > > Best regards!

---

> > > > > ### Comment · Reviewer_mBVh · 2024-08-10
> > > > >
> > > > > I read your paper again and still cannot recommend its acceptance.
> > > > >
> > > > > I still find your paper inaccessible. In my opinion, for a premier conference such as NeuRIPS, it is your responsibility to make your results accessible. Since NeuRIPS does not include a revision cycle, the submitted version of the paper must already be of high quality.
> > > > >
> > > > > * The problem addressed in the paper should be well explained and in your case, the model (Equation 1) should be justified and limitations of the model should be discussed. For instance, for fixed dimension d, the sparsity should also be a fixed value if \beta^* is the true weight parameter. Why do you only provide an upper k? When specifying the model, you do not say how k changes with d?
> > > > >
> > > > > * I also checked reference [1] "OKRidge: Scalable Optimal k-Sparse Ridge Regression", which is, in my opinion, much more accessible. In reference [1] the matrix X is not restricted to a matrix with random entries.  Such matrices are only used in the experimental section of reference [1]. So I am not sure how general your error analysis is.
> > > > >
> > > > > * The nine-page version of your paper does not contain proofs of your results, but the whole Page 4 is spent on the formal definition of GMT admissible sequences and a restatement of a theorem from reference [11]. However, the formal results are not needed to gain an informal understanding of your results.
> > > > >
> > > > >
> > > > > I am not going to change my score, but I am happy to defer the decision to an area chair.

---

> ### Author Response · Authors · 2024-08-10
>
> Dear Reviewer mBVh,
>
> Thank you for rechecking our paper and your prompt response. We apologize for any inconvenience caused by our initial presentation. To this end, following your comments, we will correct our work in the revision.
>
> In regards to your Questions
>
> __Question 1.__ The problem addressed in the paper should be well explained and in your case, the model (Equation 1) should be justified and limitations of the model should be discussed. For instance, for fixed dimension d, the sparsity should also be a fixed value if \beta^* is the true weight parameter. Why do you only provide an upper k? When specifying the model, you do not say how k changes with d?
>
> __Answer:__
> We appreciate your comment. As you referenced [1], "OKRidge: Scalable Optimal k-Sparse Ridge Regression," [1] focuses on solving the following k-Sparse Ridge Regression Optimization (k-SRO):
>
> $$\mathop{\min}\limits_ {\pmb{\beta}}\Vert \pmb{y} - \pmb{X} \pmb{\beta}\Vert_ 2^2 + \lambda \Vert\pmb{\beta}\Vert_ 2^2,   \text{s.t.} \Vert\pmb{\beta} \Vert_ 0 \leq k,$$
>
> where [1] only provides an upper bound k for sparsity. To maintain consistency with [1], we also provide an upper bound k for sparsity of the model (Equation 1).
>
> Furthermore, our paper also argues that sparsity is inherently a fixed value when $\beta^*$ is the true weight parameter. It is worth emphasizing that, regardless of the value of sparsity, it does not impact our theoretical results. Therefore, it is unnecessary to discuss how k changes.
>
> __Question 2.__ I also checked reference [1] "OKRidge: Scalable Optimal k-Sparse Ridge Regression", which is, in my opinion, much more accessible. In reference [1] the matrix X is not restricted to a matrix with random entries. Such matrices are only used in the experimental section of reference [1]. So I am not sure how general your error analysis is.
>
> __Answer:__
> To the best of our knowledge, we are the first to investigate the estimation error of the OKRidge method. The primary contribution of our paper is to provide a theoretical perspective on the estimation error of the OKRidge method under Gaussian settings.
>
> It is worth emphasizing that the Gaussian setting is a commonly used approach for theoretical analysis of algorithms in machine learning, as evidenced by papers [B1~B8]. Thus, the matrix X in our paper is restricted to Gaussian assumption. Reviewer 8imU also mentions this limitation but acknowledges our contribution under Gaussian settings. Extending our analysis to general conditions is beyond the scope of this paper and we will investigate the error of OKRidge under general conditions in our future work.
>
> __Question 3.__ The nine-page version of your paper does not contain proofs of your results, but the whole Page 4 is spent on the formal definition of GMT admissible sequences and a restatement of a theorem from reference [11]. However, the formal results are not needed to gain an informal understanding of your results.
>
> __Answer:__
> Our proofs are detailed in Appendix B~E (see lines 398-450). We have also indicated where the proofs for the key steps are located in the original paper. Additionally, the formal definition of GMT admissible sequences and the CGMT theorem from reference [11] are crucial for the derivations that follow in our manuscript. Therefore, it is important to introduce these definitions. In response to your advice, we will simplify these contents of Page 4 in our revision.
>
> Given the need for formality, rigor, and accuracy in theoretical work, it is necessary for our purely theoretical paper to provide the formal results. The informal understanding of our results is discussed in our remark 5.3 and conclusion (see lines 270-277 and 297-304).
>
> Most of your concerns pertain to our presentation, which do not impact the validity of our theoretical results. We hope our responses address your concerns and positively influence your evaluation. We are looking forward to hearing from you soon.
>
>
> [B1] Explicit Regularisation in Gaussian Noise Injections. NeurIPS 2020
>
> [B2] Communication-Constrained Bandits under Additive Gaussian Noise. ICML 2023
>
> [B3] Some Constructions of Private, Efficient, and Optimal K-Norm and Elliptic Gaussian Noise. COLT 2024
>
> [B4] Asymmetric Heavy Tails and Implicit Bias in Gaussian Noise Injections. ICML 2021
>
> [B5] Pitfalls of Gaussians as a noise distribution in NCE. ICLR 2023
>
> [B6] Precise Error Analysis of Regularized M-Estimators in High Dimensions. IEEE Trans. Inf. Theory. 2018
>
> [B7] On the Properties of Kullback-Leibler Divergence Between Multivariate Gaussian Distributions. NeurIPS 2023
>
> [B8] Hyperbolic VAE via Latent Gaussian Distributions. NeurIPS 2023

---

> > ### Comment · Reviewer_mBVh · 2024-08-12
> > **k as a function of d?**
> >
> > I do not understand your answer to Question 1.
> >
> > Note that in reference [1] the optimization problem is given, but not the model. Of course, when you solve a concrete sparse ridge regression problem, then k should be fixed and, in the optimization problem, it is enough to use k as an upper bound on the sparsity, zero-norm of $\beta*$. But you specify a model that you are analyzing in a high-dimensional setting, where $\lim_{d\rightarrow \infty} \frac{n(d)}{d}$ converges to a constant. So, in any fixed dimension $d$, the true parameter vector should have a definite sparsity value $k(d)$. Therefore, in the specification of the model, the sparsity should be given by an equality, and not by an inequality. Moreover, such as $n$, the number of data points, the sparsity should be a function of $d$. Otherwise, $k$ would be a dimension-independent constant, which makes no sense to me.  What could make sense is that $\frac{k(d)}{d}$ is constant, and maybe that is your assumption. Actually, I would be very surprised if the sparsity value does not impact your theoretical analysis. When the sparsity is $k(d)=d$, then the ridge regression problem can be solved exactly and there is no error for any value of $\lambda$. So the function $\Delta (\lambda)$ should depend on $k(d)$.

---

> ### Author Response · Authors · 2024-08-13
> **k is a function of d**
>
> Dear Reviewer mBVh:
>
> We sincerely appreciate the time and effort you have invested in the discussion period. We sincerely apologize for our misunderstanding of the concepts of sparsity and $k$ in your Question 1. Your detailed explanation of Question 1 is valuable for us in addressing your concerns.
>
> The linear model of our paper is
> $$ \pmb{y}=\pmb{X} \pmb{\beta}^* + \pmb{\epsilon} \;\text{ with }\; \Vert\pmb{\beta}^* \Vert_0 \leq k, \tag{1}$$
>
> where $\pmb{\beta}^*\in\mathbb{R}^d$ represents the ``true" weight parameter. We provide an upper $k$ to maintain consistency with the forms of the k-Sparse Ridge Regression Optimization (k-SRO):
> $$\mathop{\min}\limits_ {\pmb{\beta}}\Vert \pmb{y} - \pmb{X} \pmb{\beta}\Vert_ 2^2 + \lambda \Vert\pmb{\beta}\Vert_ 2^2,   \text{s.t.} \Vert\pmb{\beta} \Vert_ 0 \leq k. \tag{2}$$
>
> Our paper addresses the worst-case scenario $\Vert\pmb{\beta}^* \Vert_0 = k$ for the linear model (1), we apologize for any inconvenience caused by our presentations and we will further clarify this in our revision.
>
> In our paper, $k$ is indeed a function of $d$, as we assume $\frac{D(\tau)}{n}\to \bar{D}(\tau)\in (0,1)$ as $d\to \infty$, where $D(\tau)$ is related to $k$ (see lines 253 and 266). Following your suggestion, we will further emphasize this in our revision. Under this assumption, $k(d)/d$ can be a constant.
>
> To illustrate the impact of sparsity value on our theoretical results, the sparsity value $k/d$ should be a constant once model (1) is selected, which means that $k$ changes proportionally with $d$. For ease of understanding, $k/d$ and $n/d$ can be regarded as the inherent attributes of model (1); they become known once model (1) is selected. For any given model (1), when we apply the OKRidge method to estimate $\pmb{\beta}^*$, under the assumption  $\frac{D(\tau)}{n}\to \bar{D}(\tau)\in (0,1)$, we arrive at the theoretical result
> $$\lim_ {d\to \infty}\lim_ {\sigma\to 0}\text{NSE}\stackrel{P}{\longrightarrow}\Delta(\hat{\lambda}).\tag{6}$$
>
> In other words, our theoretical result (6) holds for any selected sparsity value $k/d$. This is why I assert that the sparsity value (i.e., the choice of model (1)) does not impact our theoretical results. This assertion does not imply that " When the sparsity is $k(d)=d$, the ridge regression problem can be solved exactly and there is no error for any value of $\lambda$", because our theoretical results only hold in probability for the linear model (1) under the conditions $\sigma\to 0$ and $d\to \infty$.
>
>
> Based on the analysis above, once model (1) is selected, $\Delta(\hat{\lambda})$ depends on $\lambda$. This is not contradictory to the notion that $\Delta(\hat{\lambda})$ should depend on sparsity $k/d$, because $\Delta(\hat{\lambda})$ is also influenced by the selection of a model (1) which is determined by $k(d)$ and $n(d)$. However, since our paper assumes that model (1) has already been selected, the sparsity $k/d$ is a known constant, and therefore $\Delta(\hat{\lambda})$ depends on $\lambda$.
>
>
> We would like to once again express our sincere gratitude for the time, effort, and careful consideration you have dedicated to our discussions.  Engaging with you has been invaluable for our growth. We hope our responses have met your expectations and satisfaction. We kindly request the opportunity to address any additional concerns you may have. Thank you for your time and attention to our work.  We are looking forward to your response soon.

---

### Official Review · Reviewer_uvCX · 2024-07-11

**Soundness:** 3
**Presentation:** 3
**Contribution:** 3
**Rating:** 8
**Confidence:** 4

**Summary:**

The authors provide a theoretical error analysis for the OKRidge method, which is both faster and more accurate than existing approaches for solving sparse ridge regression. The experimental results are in excellent agreement with the theoretical findings.

**Strengths:**

1. The authors analyze the estimation error of OKRidge using Convex Gaussian min-max theorem, and the proof appears to be correct.

2. The theoretical and experimental results validate the reliability of the OKRidge method.

3. This theoretical error analysis provides support for the broad application of OKRidge.

**Weaknesses:**

1. Because the OKRidge method is used to solve a k-sparse linear regression problem and the authors make the Gaussian assumption, the following related literatures are recommended to be cited:

> * 1.1 Mohsen Bayati and Andrea Montanari. The lasso risk for Gaussian matrices. Information Theory, IEEE Transactions on, 58(4):1997–2017, 2012

> * 1.2 D. Bertsimas, J. Pauphilet, and B. Van Parys. Sparse regression: Scalable algorithms and empirical performance. Statistical Science, 35(4):555–578, 2020

> *1.3 Peter J Bickel, Yaacov Ritov, and Alexandre B Tsybakov. Simultaneous analysis of lasso and dantzig selector. The Annals of Statistics, 37(4):1705–1732, 2009.

2. The authors provide numerical experiments with $n/d=0.4, 0.5 and 0.6$, but the sparsity is just $k/d=0.1$. I want to see more experiments with various sparsity.

**Questions:**

See the weaknesses.

**Limitations:**

The authors adequately addressed the limitations and potential negative societal impact of their work.

---

> ### Author Rebuttal · Authors · 2024-08-05
>
> ## Answer to Reviewer uvCX
>
> Dear Reviewer uvCX,
>
> Thank you very much for your detailed and thorough review of our paper. We sincerely appreciate the time and effort you have dedicated to providing insightful comments and bringing these issues to our attention.
>
> ### In regards to your Weaknesses:
>
>
> __Weakness 1.__  Because the OKRidge method is used to solve a k-sparse linear regression problem and the authors make the Gaussian assumption, the following related literatures are recommended to be cited:
>
> 1.1 Mohsen Bayati and Andrea Montanari. The lasso risk for Gaussian matrices. Information Theory, IEEE Transactions on, 58(4):1997–2017, 2012
>
> 1.2 D. Bertsimas, J. Pauphilet, and B. Van Parys. Sparse regression: Scalable algorithms and empirical performance. Statistical Science, 35(4):555–578, 2020
>
> 1.3 Peter J Bickel, Yaacov Ritov, and Alexandre B Tsybakov. Simultaneous analysis of lasso and dantzig selector. The Annals of Statistics, 37(4):1705–1732, 2009.
>
> __Answer:__ Thank you for you suggestions. Following your suggestions, we will cite these related literatures in our revision.
>
>
> __Weakness 2.__  The authors provide numerical experiments with $n/d=0.4, 0.5$ and $0.6$, but the sparsity is just $k/d=0.1$. I want to see more experiments with various sparsity.
>
> __Answer:__ Thank you for your suggestion. Following your suggestion, we conduct experiments with sparsity levels $0.1, 0.15,\text{and}\ 0.2$. The experimental results can be seen in the pdf of Author Rebuttal by Authors.
>
> The experimental findings with various sparsity levels demonstrate that the NSE converges to a fixed constant determined by $\lambda$, aligning excellently with our theoretical predictions. We will add these experimental results in our revision.

---

> > ### Comment · Reviewer_uvCX · 2024-08-09
> >
> > Thanks for the clarification, I'll update my score.

---

> > > ### Author Response · Authors · 2024-08-11
> > > **Appreciation for Raising Score**
> > >
> > > We sincerely thank you for recognizing our work and raising our score.

---

### Official Review · Reviewer_mn2F · 2024-07-11

**Soundness:** 3
**Presentation:** 3
**Contribution:** 4
**Rating:** 8
**Confidence:** 4

**Summary:**

OKRidge proposed in [1] shows promising results and becomes the SOTA sparse ridge regression solvers. This paper conducts the first error analysis for OKRidge. Convex Gaussian min-max theorem (CGMT) is well introduced in this paper. Based on CGMT, they develop the asymptotic theory for OKRidge. The experiments are also conducted to support the theory. This is a solid and up-to-date work.

**Strengths:**

The paper first uses CGMT to conduct the rigorous analysis for OKRidge, which is an up-to-date research. It is novel to reformulate the estimation error of OKRidge as a PO problem and further reduce the PO problem to an AO problem. By analyzing the AO problem, they show that the NSE of OKRidge tends to a constant. Then the estimators of OKRidge is close to the groundtruth. The empirical results also support the theories presented in this paper. The theoretical results in this paper are original which has never been explored before. This paper bridges the gap between theory and practice.

**Weaknesses:**

The paper presents both GMT and CGMT. But the difference between these two concepts and the advantages of CGMT over GMT are not well clarified.

In Line 208, the LHS of (23) misses the \min_{\beta}.

It is interesting to reduce the AO problem (26) to an equivalent optimization (35) that only involves two scalar variables. But more descriptions about why can we do this transformation are needed.

In Line 300, it is improper for the formulation to use “=”, as it should indicate an approximation in probability according to Theorem 5.2.

**Questions:**

1, What is the difference between GMT and CGMT and what are the advantages of CGMT over GMT?
2, Why can we reduce the problem (26) to (35) that only involves two scalar variables?

**Limitations:**

The author has presented the limitations in the checklist.

---

> ### Author Rebuttal · Authors · 2024-08-05
>
> ## Answer to Reviewer mn2F
>
> Dear Reviewer mn2F,
>
> Thank you for your job in reviewing our paper. We are very sorry for the inconvenience caused by our presentations. To this end, following your comments, we will correct our work in the revision.
>
> ### In regards to your Weaknesses:
>
> __Weakness 1.__ The paper presents both GMT and CGMT. But the difference between these two concepts and the advantages of CGMT over GMT are not well clarified.
>
> __Answer:__ We have introduced the concepts of GMT and CGMT, and clarified the difference between GMT and CGMT in Section 3.2:
>
> ``CGMT originates from Gordon’s Gaussian Min-max Theorem (GMT), which provides probabilistic bounds on the optimal cost of $\textbf{PO}$ problem via a simpler $\textbf{AO}$ problem. CGMT further applies convexity assumptions to tighten the upper and lower bounds of both the optimal cost and the norm of the optimal solution of the original problem.''
>
> The CGMT framework has been utilized to analyze the performance of solutions to non-smooth regularized convex optimization problems, which inspires us to apply the CGMT framework to analyze the NSE of the OKRidge method.
>
> (Please refer to Section 3.2 of our main paper.)
>
> __Weakness 2.__ In Line 208, the LHS of (23) misses the $\min_{\beta}$.
>
> __Answer:__ Formulation (23) is the intermediate process of deriving from (22) to (24), which is correct and doesn't affect (24). More specifically,
>
> $$\mathop{\min}\limits_ {\pmb{\beta}}\frac{1}{\sqrt{n}}\big[\Vert \pmb{X} (\pmb{\beta}-\pmb{\beta}^*) + \pmb{\epsilon} \Vert_ 2^2 + \lambda\text{SumTop}_ {k}(\pmb{\beta}\odot\pmb{\beta})\big]\tag{22}$$
>
> We introduce the new variable $\pmb{w}:=\pmb{\beta}-\pmb{\beta}^*$ and apply the Fenchel-Moreau theorem (14) to formulation (22),
>
> $$\begin{align*}
> &\mathop{\min}\limits_ {\pmb{\beta}}\frac{1}{\sqrt{n}}\big[\Vert \pmb{X} (\pmb{\beta}-\pmb{\beta}^*) + \pmb{\epsilon} \Vert_ 2^2 + \lambda\text{SumTop}_ {k}(\pmb{\beta}\odot\pmb{\beta})\big],\\\\
> =&\mathop{\min}\limits_ {\pmb{\beta}}\frac{1}{\sqrt{n}}\big[\Vert \pmb{X} \pmb{w} - \pmb{\epsilon} \Vert_ 2^2 + \lambda\text{SumTop}_ {k}\big((\pmb{w}+\pmb{\beta}^*)\odot(\pmb{w}+\pmb{\beta}^*)\big)\big],\\\\
> =&\mathop{\min}\limits_ {\pmb{w}}\max_ {\pmb{u}} \frac{1}{\sqrt{n}}\Big[\pmb{u}^\top\pmb{X}\pmb{w}- \pmb{u}^\top \pmb{\epsilon}-\frac{\Vert\pmb{u}\Vert^2_ 2}{4}+ \lambda\text{SumTop}_ {k}\big((\pmb{w}+\pmb{\beta}^*)\odot(\pmb{w}+\pmb{\beta}^*)\big)\Big]=:\Phi_ {\text{OKRidge}}(\pmb{X}),\tag{24}
> \end{align*}$$
>
> where,
> $$\begin{align*}
> &\frac{1}{\sqrt{n}}\big[\Vert \pmb{X} \pmb{w} - \pmb{\epsilon} \Vert_ 2^2 + \lambda\text{SumTop}_ {k}\big((\pmb{w}+\pmb{\beta}^*)\odot(\pmb{w}+\pmb{\beta}^*)\big)\big],\\\\
> =&\max_ {\pmb{u}} \frac{1}{\sqrt{n}}\Big[\pmb{u}^\top\pmb{X}\pmb{w}- \pmb{u}^\top \pmb{\epsilon}-\frac{\Vert\pmb{u}\Vert^2_ 2}{4}+ \lambda\text{SumTop}_ {k}\big((\pmb{w}+\pmb{\beta}^*)\odot(\pmb{w}+\pmb{\beta}^*)\big)\Big],\tag{23}
> \end{align*}$$
>
> Therefore, the formulation (23) doesn't need ``$\mathop{\min}\limits_ {\pmb{\beta}}$''.
>
> (Please refer to lines 206-212 of our main paper.)
>
>
> __Weakness 3.__ It is interesting to reduce the AO problem (26) to an equivalent optimization (35) that only involves two scalar variables. But more descriptions about why can we do this transformation are needed.
>
> __Answer:__ Theorem 3.2 indicates that, if the optimal cost $\phi(\pmb{g},\pmb{h})$ of $\textbf{AO}$ concentrates to some value $\mu$, the same holds true for $\Phi(\pmb{G})$ of $\textbf{PO}$. Furthermore, under appropriate additional assumptions, the optimal solutions of the $\textbf{AO}$ and $\textbf{PO}$ problems are also closely related by $\Vert\pmb{w}_ {\Phi}(\pmb{G})\Vert = \Vert \pmb{w}_ {\phi}(\pmb{g},\pmb{h})\Vert$, as $n\to \infty$. This suggests that, within the CGMT framework, a challenging $\textbf{PO}$ problem can be replaced with a simplified $\textbf{AO}$ problem, from which the optimal solution of the $\textbf{PO}$ problem can be accurately inferred. Moreover, we are more concerned about $\Vert \pmb{w}_ {\phi}(\pmb{g},\pmb{h})\Vert$. Therefore, if we reduce the AO problem only involves scalar variable about $\Vert \pmb{w}_ {\phi}(\pmb{g},\pmb{h})\Vert$, we obatain the error of OKRidge method by the relationship $\Vert\pmb{w}_ {\Phi}(\pmb{G})\Vert = \Vert \pmb{w}_ {\phi}(\pmb{g},\pmb{h})\Vert$.
>
> If the optimal solution of optimization (35) is $\alpha=\alpha^*$, we have $\Vert\pmb{w}_ {\hat{\phi}_ {\text{OKRidge}}}\Vert_ 2\stackrel{P}{\longrightarrow}\alpha^*$ for approximated $\textbf{AO}$ problem (28). If $\alpha^* $ further tends to $0$, according to formulation (29) and CGMT, $\Vert\pmb{w}_ {\Phi_{\text{OKRidge}}}\Vert_ 2\stackrel{P}{\longrightarrow} \alpha^*$ holds for $\textbf{PO}$ problem (24). Then, for the estimation error of OKRidge produced by (21), we have $\Vert\hat{\pmb{\beta}}-\pmb{\beta}^*\Vert_ 2\stackrel{P}{\longrightarrow} \alpha^* $. Therefore, it only remains to obtain the optimal value of $\alpha$ in optimization (35) that plays the role of $\Vert\pmb{w}\Vert_ 2$.
>
> (Please refer to lines 152-158 and 259-264 of our main paper.)
>
>
> __Weakness 4.__ In Line 300, it is improper for the formulation to use “=”, as it should indicate an approximation in probability according to Theorem 5.2.
>
> __Answer:__ Thank you for your comments. We will change the “=” to ``$\stackrel{P}{\longrightarrow}$'' in our revision.
>
> ### In regards to your Questions:
>
> __Question 1.__ What is the difference between GMT and CGMT and what are the advantages of CGMT over GMT?
>
> __Answer:__ Thank you for your comments. Question 1 is similar to Weakness 1 . Please see our Answers to Weakness 1.
>
> __Question 2.__ Why can we reduce the problem (26) to (35) that only involves two scalar variables?
>
> __Answer:__ Thank you for your comments. Question 2 is similar to Weakness 3. Please see our Answers to Weakness 3.

---

> > ### Comment · Reviewer_mn2F · 2024-08-09
> >
> > Thanks to the authors for the detailed responses. You have solved most of my problems, thanks.

---

> > > ### Author Response · Authors · 2024-08-09
> > > **Appreciation to Reviewer mn2F**
> > >
> > > Thank you very much for your acknowledgment and efforts.

---

### Official Review · Reviewer_553k · 2024-07-11

**Soundness:** 2
**Presentation:** 3
**Contribution:** 2
**Rating:** 4
**Confidence:** 2

**Summary:**

This paper improves the theoretical reliability of OKRidge method for the sparse ridge regression by introducing a theoretical error analysis. OKRidge is reframed in this paper as a Primary Optimization problem. Then this paper use the Convex Gaussian Min-max Theorem (CGMT) to simplify it to an Auxiliary Optimization problem. This paper provide a theoretical error analysis for OKRidge based on the  problem.

**Strengths:**

- The paper provides a novel theoretical error analysis for the OKRidge method, leveraging the CGMT framework, which is a contribution to the field of sparse regression.
- The paper is well-structured, with a clear outline of the problem, methodology, and results.

**Weaknesses:**

- While the theoretical contributions are clear, the practical motivation for why this particular error analysis is crucial could be better articulated. More emphasis on how this analysis directly impacts real-world applications would strengthen the paper.
- The experiments, although validating the theoretical claims, are limited in scope. They primarily focus on synthetic data with Gaussian noise. Including more diverse datasets, especially real-world examples, would enhance the validity of the results.
- The reliance on Gaussian input settings is a significant limitation. The paper acknowledges this but does not provide a clear path for extending the results to non-Gaussian settings. This could limit the applicability of the findings.

**Questions:**

- Can the authors provide more insight into the practical applications where this theoretical error analysis will be most impactful?
- Are there any specific strategies proposed for extending the theoretical findings to non-Gaussian input settings?
- How robust are the experimental findings to variations in the underlying assumptions, such as the sparsity level and noise distribution?

**Limitations:**

The authors have addressed the limitations related to Gaussian input settings and indicated future work on extending the results to non-Gaussian settings.

---

> ### Author Rebuttal · Authors · 2024-08-05
>
> Dear Reviewer 553k, we truly appreciate the patience and effort you've dedicated to providing valuable feedback. We appreciate the opportunity to address your concerns.
>
> ### For Weaknesses:
>
> __W1:__ Error analysis of algorithms is a very crucial and popular topic in the field of machine learning, as evidenced by several papers [A1-A8]. Therefore, it is important to study the theoretical error of this advanced OKRidge algorithm which is both faster and more accurate than existing approaches [1]. In our paper, there have been some contents of practical motivation and impact on real-world applications:
>
> (1). Practical motivation:
>
> Sparse Ridge Regression (SRR) has achieved notable success in machine learning applications, including statistics [2], signal processing [3], dynamical systems [4], and others. The OKRidge method is both faster and more accurate than existing approaches in solving SRR. However, the absence of theoretical analysis on the error of OKRidge impedes its large-scale applications. We provide a theoretical perspective on the estimation error of the OKRidge method under Gaussian assumption. (see lines 1-15 and 29-39)
>
> (2). Impact on real-world applications:
>
> i). This theoretical error analysis substantiates the reliability of OKRidge and provides guidelines on the error analysis of other algorithms. (see lines 10-11 and 303-304)
>
> ii). Our analysis strengthens the theoretical underpinnings of OKRidge and provides theoretical reliability for its broad application in the real-world. (see lines 59-60)
>
> ii). Our work provides theoretical support for the broad application of OKRidge, which does not require proprietary software or expensive licenses, unlike its main competitor [1]. This can significantly impact various regression applications. (see lines 395-397)
>
> Following your suggestion, in our revision, we will provide more practical motivation and impacts on real-world applications to strengthen our paper.
>
> [A1] Estimating the Error of Randomized Newton Methods: A Bootstrap Approach. ICML 2020
>
> [A2] $l_{1, p}$-Norm Regularization: Error Bounds and Convergence Rate Analysis of First-Order Methods. ICML 2015
>
> [A3] Addressing Function Approximation Error in Actor-Critic Methods. ICML 2018
>
> [A4] Faster Algorithms and Constant Lower Bounds for the Worst-Case Expected Error. NeurIPS 2021
>
> [A5] A Comparison of Hamming Errors of Representative Variable Selection Methods. ICLR 2022
>
> [A6] On Generalization Error Bounds of Noisy Gradient Methods for Non-Convex Learning. ICLR 2020
>
> [A7] Generalization error of spectral algorithms. ICLR 2024
>
> [A8] Error Estimation for Randomized Least-Squares Algorithms via the Bootstrap. ICML 2018
>
> __W2:__ The OKRidge proposed by [1]  is both faster and more accurate than existing approaches in solving SRR problems. But, the paper [1] lacks theoretical analysis on the estimation error of OKRidge. To the best of our knowledge, we are the first to fill this gap. The primary contribution of our paper is to provide a theoretical perspective on the estimation error of the OKRidge method under Gaussian settings, where the Gaussian hypothesis is a commonly used approach in learning theory, as evidenced by papers [B1~B6]. Our paper is purely theoretical and Reviewer uvCX verifies the correctness of our theorems. The numerical experiments in our paper are sufficient to validate our theoretical claims. Moreover, the experiments of OKRidge on real-world examples have been done by [1] (See Figure 3 in [1]), which shows that the error of OKRidge tends to 0. This verifies the validity of our theorems.
>
> [B1] Explicit Regularisation in Gaussian Noise Injections. NeurIPS 2020
>
> [B2] Communication-Constrained Bandits under Additive Gaussian Noise. ICML 2023
>
> [B3] Some Constructions of Private, Efficient, and Optimal K-Norm and Elliptic Gaussian Noise. COLT 2024
>
> [B4] Asymmetric Heavy Tails and Implicit Bias in Gaussian Noise Injections. ICML 2021
>
> [B5] Pitfalls of Gaussians as a noise distribution in NCE. ICLR 2023
>
> [B6] Precise Error Analysis of Regularized M-Estimators in High Dimensions. IEEE Trans. Inf. Theory. 2018
>
> [B7] On the Properties of Kullback-Leibler Divergence Between Multivariate Gaussian Distributions. NeurIPS 2023
>
> [B8] Hyperbolic VAE via Latent Gaussian Distributions. NeurIPS 2023
>
> __W3:__ It is worth emphasizing that the Gaussian setting is a commonly used approach for theoretical analysis of algorithms in machine learning, as evidenced by papers [B1~B8]. Thus, our paper provides a theoretical perspective on the error of the OKRidge method under Gaussian settings. Reviewer 8imU also mentions this limitation but acknowledges our contribution under Gaussian settings. Although extending the results to non-Gaussian distributions is beyond the scope of this paper, we briefly discuss the strategies for non-Gaussian distributions here: We can utilize Fisher transformation, Box-Cox transformation, or inversion sampling to transform non-Gaussian distribution to Gaussian distribution.
>
> ### For Questions:
>
> __Q1:__ See Answer to W1.
>
> __Q2:__ See Answer to W3.
>
> __Q3:__ We analyze the error of the OKRidge method under Gaussian assumption, where the Gaussian noise level is $\sigma$. The experiments on noise level have been shown in Figures 1 and F1~F4 (a) of our paper.
>
> Following your advice, we conduct experiments on the variations in the sparsity level and noise distribution. The experimental results are shown in the pdf of Author Rebuttal by Authors. The experimental findings to variations about the sparsity level and noise distribution show that the NSE converges to a constant decided by $\lambda$, aligning excellently with our theoretical predictions. In other words, these experimental findings are robust to support our theoretical claims. We will add these experimental results in our revision.
>
> We sincerely thank you once again for your time, effort, and expertise in reviewing our manuscript. We hope our responses have met your expectations and satisfaction.

---

> > ### Author Response · Authors · 2024-08-12
> > **Clarification once again**
> >
> > Dear Reviewer 553k,
> >
> >  We sincerely appreciate the time and effort you have invested in reviewing our paper. Your concerns center around the experimental results on real-world examples and the limitation of Gaussian settings.
> >
> > (1). The comprehensive experiments of OKRidge on real-world examples were conducted by the NeurIPS 2023 paper [1] (see Figure 3 and Appendix H in [1]), which demonstrates that the error of OKRidge tends to zero. Our paper aims to offer a theoretical guarantee for this experimental phenomenon observed in [1]. Therefore, the experiments in [1] on real-world examples are adequate in validating the validity of our theorems, making it redundant to repeat these experiments to observe the same phenomenon again.
> >
> >
> > (2). The Gaussian setting is a widely adopted framework for theoretical analysis of algorithms in machine learning, as evidenced by papers [B1~B8]. Consequently, our work offers a theoretical perspective on the error of the OKRidge method under Gaussian settings. We also acknowledge this Gaussian setting in our Limitations Section (see lines 391-397). According to NeurIPS official principles and NeurIPS Reviewer principles, our limitation should not be punished, where these principles are presented by following:
> >
> > NeurIPS official principles emphsize: "The authors are encouraged to create a separate Limitations section in their paper. We understand that authors might fear that complete honesty about limitations might be used by reviewers as grounds for rejection. Reviewers will be specifically instructed to not penalize honesty concerning limitations." (see NeurIPS official website)
> >
> > NeurIPS Reviewer principles emphsize:" Authors should be rewarded rather than punished for being upfront about the limitations of their work and any potential negative societal impact." (see the Limitations part of NeurIPS Reviewer operation panel)
> >
> > Thank you once again for your consideration and assistance. We understand that you have other commitments, and we apologize for any disruption this follow-up may cause. Since there are only a few days left in the discussion period, your timely feedback would be greatly valued and essential to our process. We look forward to hearing from you soon.
> >
> >
> > [1] Okridge: Scalable optimal k-sparse ridge regression for learning dynamical systems. In NeurIPS, 2023.

---

> > > ### Comment · Area_Chair_fmWz · 2024-08-12
> > >
> > > Dear Reviewer 553k:
> > >
> > > Can you please respond to the rebuttal as soon as possible? Your comments will be greatly appreciated. Many thanks,
> > >
> > > AC

---

> ### Author Response · Authors · 2024-08-10
> **Kindly Requesting Confirmation on Responses**
>
> Dear Reviewer 553k,
>
> We hope this message finds you well. I am writing to kindly follow up on our previous correspondence and to request your feedback regarding whether our responses have adequately addressed your concerns. We sincerely appreciate the time and effort you have invested in reviewing our paper.
>
> We understand that you have other commitments, and we apologize for any inconvenience this follow-up may cause. Your timely feedback would be greatly valued and is essential for us to move forward.
>
> Thank you once again for your consideration and assistance. We look forward to hearing from you soon.
>
> Best regards!

---

### Author Rebuttal · Authors · 2024-08-05

Additional experiments on the variations about the sparsity level and noise distribution can be seen in following pdf.

---

### Decision · Program_Chairs · 2024-09-25

**Decision:**

Accept (poster)

**Comment:**

This paper considers a $k$-sparse linear regression model where the number of active (non-zero) coefficients is bounded by the constant $k$. The basic solution method is framed as ridge regression with the $\ell_0$ constraint. As it is NP-hard, one approach for scalable estimation is to consider a lower bound to the the objective function, as adopted in the OKRidge paper (Liu et al., NeurIPS). The contributions of this paper are theoretical (i.e., to provide theoretical justifications for OKRidge) and include the following: (i) a proposal of another novel tight lower bound to the NP-hard objective function, using the Convex Gaussian min-max theorem (CCMT);(ii) asymptotic analysis of the estimation error of OKRidge. This paper provides valuable contributions to the literature and potentially helps the practitioners to use OKRidge in large-scale applications. Throughout the author-reviewer discussion period, there were relatively active discussions and it would strengthen the paper substantially if the authors revise the paper accordingly. In addtion, the following aspects might be useful: (i) to make this paper self-contrained, it might be useful to provide a complete algorithm of OKRidge, including how to choose hyper-parameters; (ii) to streamline the exposition in sections 5.1 and 5.2 to help the readers; (iii) to more fully discuss the implications of the theoretical findings.